# Accelerating Regression Tasks with Quantum Algorithms

**Chenghua Liu** [1 2]  **Zhengfeng Ji** [3]

## Abstract

Regression is a cornerstone of statistics and machine learning, with applications spanning science, engineering, and economics. While quantum algorithms for regression have attracted considerable attention, most existing work has focused on linear regression, leaving many more complex yet practically important variants unexplored. In this work, we present a unified quantum framework for accelerating a broad class of regression tasks—including linear and multiple regression, Lasso, Ridge, Huber, $\ell_p$-, and $\gamma_p$-type regressions—achieving up to a quadratic improvement in the number of samples $m$ over the best classical algorithms. This speedup is achieved by a non-trivial quantization of the recent classical breakthrough of Jambulapati et al. (2024), where we construct a full quantum pipeline that strategically employs quantum leverage score approximation to initialize and refine Multiscale Leverage Score Overestimates, enabling efficient importance sampling via the preparation of multiple state copies. For problems of dimension $n$, sparsity $r < n$, and error parameter $\epsilon$, our algorithm solves the problem in $\widetilde{O}(r\sqrt{mn}/\epsilon + \text{poly}(n, 1/\epsilon))$[1] quantum time, demonstrating both the applicability and the efficiency of quantum computing in accelerating regression tasks.

## 1. Introduction

Regression lies at the heart of statistical modeling and data analysis, providing a fundamental framework for under-standing relationships between variables and making predictions. Among its various forms, linear regression stands out as one of the most widely used and longest-standing tools, with a history dating back over three centuries and applications ranging from the natural sciences to economics and engineering (Legendre, 1806; Fisher, 1992; Neter et al., 1996). Over the decades, the basic linear regression model has been extended in multiple directions, including ridge and lasso regression, $\ell_p$ regression, Huber regression, each addressing different statistical and computational challenges. These models have become indispensable in modern data analysis, driving extensive theoretical research on fast regression methods. Among them, $\ell_p$ regression has witnessed a series of recent breakthroughs (Adil et al., 2019; Lee et al., 2019; Musco et al., 2022; Adil et al., 2024; Jambulapati et al., 2024), alongside many other advances in other regression tasks.

With the rapid progress of quantum hardware and algorithms, there has been growing interest in leveraging quantum computing to accelerate optimization and machine learning tasks. Among them, regression stands out as a core problem in statistics and machine learning, and has attracted significant attention from the quantum algorithms community. A rich body of work has explored quantum approaches to linear regression under various models and assumptions, ranging from HHL-based solvers and the conditional gradient method to quantum leverage score techniques (Wiebe et al., 2012; Schuld et al., 2016; Wang, 2017; Kerenidis & Prakash, 2017; Chakraborty et al., 2019; Shao & Xiang, 2020; Chen & de Wolf, 2023; Shao, 2023; Chen et al., 2023; Chakraborty et al., 2023; Song et al., 2023). However, for broader and both theoretically and practically important regression tasks such as $\ell_p$ regression and Huber regression, it remains unclear whether similar quantum speedups can be achieved. Given the diversity of regression tasks and their substantial significance in both machine learning and theoretical computer science, a natural question arises:

*To what extent can the advantages of quantum computing be extended beyond linear regression to encompass this broader family of models?*

In this work, we study this question by adopting the generalized linear model (GLM) sparsification framework and developing a quantum algorithmic approach tailored to it. Spar-

---

[1]Key Laboratory of System Software (Chinese Academy of Sciences), Institute of Software, Chinese Academy of Sciences, Beijing, China [2]University of Chinese Academy of Sciences, Beijing, China [3]Department of Computer Science and Technology, Tsinghua University, Beijing, China. Correspondence to: Zhengfeng Ji <jizhengfeng@tsinghua.edu.cn>.

*Proceedings of the 43rd International Conference on Machine Learning*, Seoul, South Korea. PMLR 306, 2026. Copyright 2026 by the author(s).

[1]We use $\widetilde{O}(f)$ to represent $O(f \cdot \text{polylog}(m, n, 1/\epsilon))$ throughout this paper to suppress polylogarithmic factors.

sification refers to the process of replacing a large dataset with a much smaller weighted subset (a sparsifier) that approximately preserves the objective value for all candidate solutions, thereby allowing the original optimization problem to be solved more efficiently. Once such a sparsifier is constructed, computations performed on it yield an approximate solution to the original problem. Consequently, our framework provides quantum speedups for approximately solving a broad class of empirical risk minimization (ERM) problems, covering linear and multiple regression, Lasso, ridge, Huber, $\ell_p$-, and $\delta_p$-type regressions. This demonstrates both the unifying nature and the broad applicability of our approach in accelerating fundamental regression tasks.

To clearly specify the scope of our framework, we formalize the problem of optimizing a generalized linear model as

$$\min_{x \in \mathbb{R}^n} F(x) \quad \text{for} \quad F(x) := \sum_{i=1}^m f_i\big(\langle a_i, x \rangle - b_i\big), \quad (1)$$

where the total loss $F : \mathbb{R}^n \to \mathbb{R}$, is defined by data vectors $a_1, \ldots, a_m \in \mathbb{R}^n$, a response vector $b \in \mathbb{R}^m$, and loss functions $f_1, \ldots, f_m : \mathbb{R} \to \mathbb{R}_+$. This general formulation serves as a unifying abstraction for many regression tasks, from linear regression to more sophisticated robust objectives.

When $m \gg n$, a natural strategy for accelerating computation is to apply sparsification. This technique has been widely used in classical algorithm design, with graph sparsification being one of the most prominent examples (Spielman & Srivastava, 2011; Batson et al., 2012). In a recent breakthrough, Jambulapati et al. (2024) systematically studied the GLM sparsification problem and presented a corresponding classical algorithm. Motivated by this advance, we investigate the potential for quantum speedups in this setting, leading to our quantum algorithmic GLM sparsification framework. To precisely describe GLM sparsification, we use the following definition:

**Definition 1.1** (Generalized Linear Model (GLM) Sparsification)**.** Let the total loss function $F(x) := \sum_{i=1}^m f_i\left(\langle a_i, x \rangle - b_i\right)$ be defined by vectors $a_1, \ldots, a_m \in \mathbb{R}^n$, a vector $b \in \mathbb{R}^m$, and loss functions $f_1, \ldots, f_m : \mathbb{R} \to \mathbb{R}_+$. We say that $F$ admits an $s$-sparse $\epsilon$-approximation over the range $[s_{\min}, s_{\max}]$ if there exist non-negative weights $w \in \mathbb{R}_+^m$, with at most $s$ nonzero entries, such that

$$\left| F(x) - \widetilde{F}(x) \right| \leq \epsilon \cdot F(x)$$

for all $x \in \mathbb{R}^n$ such that $s_{\min} \leq F(x) \leq s_{\max}$. Here $\widetilde{F}(x) := \sum_{i=1}^m w_i f_i(\langle a_i, x \rangle - b_i)$ is referred to as an $s$-sparse $\epsilon$-approximate sparsifier of $F$ over $[s_{\min}, s_{\max}]$.

## 1.1. Our results

We consider a generalized linear model as in Equation (1), where the loss functions $f_1, \ldots, f_m : \mathbb{R} \to \mathbb{R}_+$ are applied to $\langle a_i, x \rangle - b_i$ for $a_i \in \mathbb{R}^n$. Throughout the analysis, we assume for convenience that the bias term is absent, i.e., $b = 0$. This assumption entails no loss of generality: any nonzero bias $b_i$ can be incorporated into the vector by appending an additional coordinate, rewriting $\langle a_i, x \rangle - b_i$ as $\langle (a_i, b_i), (x, -1) \rangle$, thereby embedding the problem in $\mathbb{R}^{n+1}$ with zero bias.

To enable effective GLM sparsification, we require that each loss function $f_i$ satisfies certain regularity conditions. Specifically, we focus on losses that are $L$-auto-Lipschitz and lower $\theta$-homogeneous—two mild and widely satisfied properties that imply several other desirable characteristics for sparsification. These conditions, along with their consequences, are discussed in detail by Jambulapati et al. (2024). Following their results, we formalize this class of losses as $(L, \theta, c)$-proper functions, defined as follows:

**Definition 1.2** (Proper Loss Functions)**.** A family of loss functions $\mathcal{F} = \{f_1, \ldots, f_m\}$ is called $(L, \theta, c)$-proper if for each $f_i : \mathbb{R} \to \mathbb{R}_+$, the transformed function $h_i = \sqrt{f_i}$ satisfies:

1. ($L$-auto-Lipschitz) $|h_i(x) - h_i(x')| \leq L h_i(x - x')$ for all $x, x' \in \mathbb{R}$.

2. (Lower $\theta$-homogeneous with $c$) $h_i(\lambda x) \geq c\lambda^\theta h_i(x)$ for all $x \in \mathbb{R}$ and $\lambda \geq 1$.

We assume $c > 0, L > 0, \theta \in (0, 4)$; in particular, $\theta = 1$ already suffices for all applications in this paper.

In this paper, we introduce a quantum algorithmic framework for GLM sparsification, capable of efficiently constructing $\epsilon$-approximate sparsifiers for a broad class of $(L, \theta, c)$-proper loss functions. Given query access to the data matrix and the loss functions, our method achieves quadratic speedups in the number of data points $m$ over the fastest classical algorithms, while preserving the same approximation guarantees. Our main result is summarized in the following theorem.

**Theorem 1.3** (Informal, see Theorem 4.1 for the formal statement)**.** *Let $F(x) := \sum_{i=1}^m f_i\left(\langle a_i, x \rangle\right)$ be a total loss function defined by vectors $a_1, \ldots, a_m \in \mathbb{R}^n$ (with sparsity $r \leq n$) and an $(L, \theta, c)$-proper loss functions family $\mathcal{F} = \{f_1, \ldots, f_m\}$. For any $\epsilon = O(1/r), s_{\max} > s_{\min} \geq 0$, there exists a quantum algorithm that, with high probability[2], constructs an $\widetilde{O}(\log(s_{\max}/s_{\min}) \cdot n/\epsilon^2)$-sparse $\epsilon$-approximate sparsifier of $F$ over the range $[s_{\min}, s_{\max}]$, in time $\widetilde{O}\left((r\sqrt{mn}/\epsilon + \mathrm{poly}(n)) \cdot \log(s_{\max}/s_{\min})\right)$.*

---

[2]Throughout this paper, we say an event holds "with high probability" (w.h.p.) if it holds with probability at least $1 - O(1/n)$.

We illustrate applicability with common loss functions. It is straightforward to verify that $\ell_p(x) = |x|^p$ for $p \in (0, 2]$ is $(L, \theta, c)$-proper with $L = 1$, $\theta = p/2$, and $c = 1$. For the more refined function $\gamma_p$ defined for $p \in (0, 2]$ by

$$\gamma_p(x) := \begin{cases} \frac{p}{2}x^2 & \text{for } |x| \leq 1, \\ |x|^p - (1 - p/2) & \text{otherwise,} \end{cases} \quad (2)$$

one similarly checks $(L, \theta, c)$-properness with the same parameters, where the case $p = 1$ recovers the Huber loss. Regarding the dependence on the scale ratio $s_{\max}/s_{\min}$, when all $f_i$ are $p$-homogeneous the dependence disappears, and for $\gamma_p$ the scale ratio reduces to $\text{poly}(m)$.

As an application of our framework, we achieve quantum speedups for a range of important regression problems, including linear regression, multiple regression, Lasso, Ridge, Huber, $\ell_p$ regression, and $\gamma_p$ regression. Since the sparsifier has size $\widetilde{O}(n/\epsilon^2)$, it must be smaller than $m$ (otherwise it would be dense), which requires $\epsilon = \Omega(\sqrt{n/m})$. In this regime, the leading term in our quantum runtime, $r\sqrt{mn}/\epsilon$, is strictly smaller than the classical $mr$; when $\epsilon$ is constant, this yields quadratic speedups in the parameter $m$, which often dominates in large-scale regression tasks where $m \gg n$. Moreover, for linear, multiple, and Ridge regression, our algorithm also subsumes the results of Song et al. (2023), the previous known quantum algorithm for the problem. Finally, we discuss quantum lower bounds in Remark 4.3, demonstrating that our algorithm achieves *near-optimal* dependence on $m$ and $\epsilon$ in relevant regimes.

We emphasize that our focus is on the computational task of approximately minimizing the loss function $F$, rather than on the statistical learning problem of estimating an unknown model from random samples. The latter setting was studied by Chen & de Wolf (2023), who investigated quantum algorithms for Lasso and ridge regression; in our notation, their results correspond to the case $m = O(\log n/\epsilon^2)$. To facilitate a direct comparison of computational complexity, Table 1 summarizes the runtimes across classical algorithms, prior quantum work, and our results.

**Concurrent work** In a concurrent and independent work, Song et al. (2025) study $\ell_p$ regression within a quantum sensitivity-sampling framework and show that one can construct an $\epsilon$-coreset of size $\widetilde{O}(n/\epsilon^2)$ in time $\widetilde{O}(n\sqrt{mn}(\epsilon^{-3} + \sqrt{n}))$.

## 1.2. Techniques

**Generalized linear model sparsification.** To systematically investigate quantum speedups for a broad class of regression problems, we adopt the generalized linear model (GLM) sparsification framework and further extend its applicability by allowing linear combinations of loss functions. At a high level, GLM sparsification replaces a large dataset

with a much smaller weighted subset (the sparsifier) such that the total loss is preserved within a $(1 \pm \epsilon)$ factor for all choices of the variable $x$. This enables the original optimization problem to be approximated efficiently by solving it on the sparsifier. Recently established in the breakthrough works of Jambulapati et al. (2023; 2024), this framework unifies and generalizes sparsification techniques for a wide range of regression objectives. Its core idea is to compute, for each loss function $f_i$, an importance score capturing its contribution to the objective, and then perform importance sampling to select a small representative set, reweighting them to form the sparsifier. In the quantum setting, our approach comprises two parts: (i) computing importance scores efficiently, and (ii) performing importance sampling based on these scores.

**Multiscale leverage score overestimates.** Defining suitable importance scores in the GLM setting is challenging due to its high level of abstraction: rather than specifying a concrete loss function, only general conditions on $f_i$ are given. Jambulapati et al. (2024) addressed this by analyzing the iterated covering argument for $\ell_p$ loss and generalizing $\ell_p$ Lewis weights (Lewis, 1979; Bourgain et al., 1989; Cohen & Peng, 2015), yielding an *approximate weight scheme* consisting of multiple sets of weights. These can be interpreted as importance scores at different scales, connected by specific contraction properties. Building on this, we summarized the key structural ideas into the notion of *multiscale leverage score overestimates* (MLSO), which serves as the GLM analogue of importance scores.

To achieve quantum speedups, we use the results of Apers & Gribling (2024) on quantum leverage score approximation as a key foundation, enabling a quantum implementation of the contractive algorithm from Jambulapati et al. (2024). This yields a quantum data structure, built with sublinear-time preprocessing, that supports efficient queries to MLSO values. Conceptually, our algorithm iteratively refines an initial coarse weight estimate over $\widetilde{O}(1)$ rounds, followed by a final quantum refinement step to produce MLSO values. To support this process, we develop a quantum procedure for constructing the initial weight vector by slightly modifying each $f_i$ and generating an appropriate starting weight, ensuring that the entire pipeline remains fully quantum.

**Quantum importance sampling.** Once quantum query access to the importance scores is available, the next step is to perform importance sampling efficiently. The main challenge is that we do not have direct access to the normalized sampling distribution, but only to an oracle returning unnormalized importance scores. To address this, we use the "preparing many copies of a quantum state" technique of Hamoudi (2022), which allows us to prepare many copies of a quantum state whose amplitudes are proportional to the

*Table 1.* Comparison of runtimes for various regression problems in classical algorithms, previous quantum algorithms, and our work. Due to space constraints, we present only simplified expressions of the time complexities for each task; detailed discussions are provided in Section 4.2. We list only the state-of-the-art results that allow direct comparison and omit data-dependent algorithms.

| Regression type | Classical | Quantum (previous) | Quantum (this work) |
|---|---|---|---|
| Linear regression | $\widetilde{O}(mr + n^3)$ 
(Nelson & Nguyên, 2013; Clarkson & Woodruff, 2017) | $\widetilde{O}(\sqrt{m}n^{1.5}/\epsilon + \text{poly}(n, 1/\epsilon))$ 
(Song et al., 2023) | $\widetilde{O}(r\sqrt{mn}/\epsilon + n^3)$ 
(Corollary 4.6) |
| Multiple regression | $\widetilde{O}(mr + N\,\text{poly}(n, 1/\epsilon))$ 
(Nelson & Nguyên, 2013; Clarkson & Woodruff, 2017) | $\widetilde{O}(\sqrt{m}n^{1.5}/\epsilon + N\,\text{poly}(n, 1/\epsilon))$ 
(Song et al., 2023) | $\widetilde{O}(r\sqrt{mn}/\epsilon + N\,\text{poly}(n, 1/\epsilon))$ 
(Corollary 4.7) |
| Ridge regression | $\widetilde{O}(mr + \text{poly}(n, 1/\epsilon))$ 
(Avron et al., 2017) | $\widetilde{O}(\sqrt{m}n^{1.5}/\epsilon + \text{poly}(n, 1/\epsilon))$ 
(Song et al., 2023) | $\widetilde{O}(r\sqrt{mn}/\epsilon + n^3)$ 
(Corollary 4.8) |
| Lasso regression | $\widetilde{O}(mn^2 + n^3)$ 
(Efron et al., 2004) | — | $\widetilde{O}(r\sqrt{mn}/\epsilon + \text{poly}(n, 1/\epsilon))$ 
(Corollary 4.9) |
| Huber regression | $\widetilde{O}(mr + \text{poly}(n, 1/\epsilon))$ 
(Jambulapati et al., 2024) | — | $\widetilde{O}(r\sqrt{mn}/\epsilon + \text{poly}(n, 1/\epsilon))$ 
(Corollary 4.5, with $p = 1$) |
| $\ell_p$ regression | $\widetilde{O}(mr + \text{poly}(n, 1/\epsilon))$ 
(Jambulapati et al., 2024) | — | $\widetilde{O}(r\sqrt{mn}/\epsilon + \text{poly}(n, 1/\epsilon))$ 
(Corollary 4.4) |
| $\gamma_p$ regression | $\widetilde{O}(mr + \text{poly}(n, 1/\epsilon))$ 
(Jambulapati et al., 2024) | — | $\widetilde{O}(r\sqrt{mn}/\epsilon + \text{poly}(n, 1/\epsilon))$ 
(Corollary 4.5) |

importance scores. This technique is a powerful and widely applicable tool, and has also been used in several other efficient quantum algorithms (see, e.g., Xue et al., 2023; Wang et al., 2024; Liu et al., 2025; Apers et al., 2025). We combine this with a quantum sum estimation procedure (Li et al., 2019) to approximate the normalization constant, enabling correct reweighting of the sampled elements. The correctness of this method follows from a careful adaptation of the guarantees established in Jambulapati et al. (2024).

## 2. Preliminaries

**Notations.** For a positive integer $n \in \mathbb{Z}_+$, let $[n] := \{1, \ldots, n\}$ and $[n]_0 := \{0, \ldots, n-1\}$. For vectors $u, v \in \mathbb{R}^n$, $\langle u, v \rangle = \sum_{i \in [n]} u_i v_i$ denotes the standard inner product. For $w \in \mathbb{R}^n$, $\text{diag}(w)$ denotes the $n \times n$ diagonal matrix with diagonal entries $w_i$.

**Quantum computational model.** We assume the usual quantum computational model, which is a classical system that can (i) run quantum subroutines on $O(\log m)$ qubits, (ii) can make quantum queries to the input, and (iii) has access to a quantum-read/classical write RAM (QRAM) of $\text{poly}(m)$ bits. The *quantum query complexity* measures the number of quantum queries that an algorithm makes. The *quantum time complexity* measures the number of elementary classical and quantum gates, quantum queries, and single-bit QRAM operations that algorithm uses. For readers less familiar with the basics of quantum computing and QRAM, we refer to Appendix B.

Regarding queries to the input, we consider an input matrix $A \in \mathbb{R}^{m \times n}$ with row sparsity $r \leq n$, where the matrix

elements can be queried via the oracle $\mathcal{O}_A^{\text{elem}} : |i\rangle|j\rangle|0\rangle \mapsto |i\rangle|j\rangle|a_{ij}\rangle, \forall i \in [m], j \in [n]$. In addition, the indices of the nonzero entries of each row can be queried via the oracle $\mathcal{O}_A^{\text{idx}} : |i\rangle|k\rangle \mapsto |i\rangle|t_{ik}\rangle, \forall i \in [m], k \in [r_i]$, where $t_{ik}$ is the column index of the $k$-th nonzero entry in the $i$-th row of $A$, and $r_i \leq r$ is the number of nonzero entries in that row. For simplicity, we use the notation $\mathcal{O}_A$ to refer to either of these two oracles. We will also use the following quantum queries: for a vector $w \in \mathbb{R}^n$, its elements can be queried via the oracle $\mathcal{O}_w : |i\rangle|0\rangle \mapsto |i\rangle|w_i\rangle, i \in [n]$; and for a family of functions $\mathcal{F} = \{f_1, \ldots, f_m\}$, we assume access to the oracle $\mathcal{O}_{\mathcal{F}} : |i\rangle|x\rangle|0\rangle \mapsto |i\rangle|x\rangle|f_i(x)\rangle$.

**Quantum leverage score approximation.** Let $A \in \mathbb{R}^{m \times n}$ be a matrix with rows $a_1, \ldots, a_m$. The *leverage score* of the $i$-th row is defined as

$$\sigma_i(A) := a_i^\top (A^\top A)^+ a_i, \quad \forall i \in [m], \qquad (3)$$

where $(A^\top A)^+$ denotes the Moore-Penrose pseudoinverse of $A^\top A$. Intuitively, $\sigma_i(A)$ measures the "statistical importance" of row $i$ with respect to the whole set of rows. Leverage scores are central to row-sampling methods that construct a smaller matrix $\widetilde{A}$ preserving the key spectral properties of $A$. Sampling rows proportionally to $\sigma_i(A)$ and rescaling yields $\widetilde{A}^\top \widetilde{A} \approx A^\top A$ with high probability, enabling accurate and efficient approximation of linear regression and other optimization tasks; we refer the reader to the survey of Woodruff (2014) for a comprehensive overview.

In a recent important work, Apers & Gribling (2024) introduced a simple recursive halving algorithm for quantum spectral approximation. As a corollary, they obtained a quantum algorithm for leverage score approximation, which

serves as a cornerstone of our work—we invoke this result multiple times throughout our algorithms. For completeness, we restate their result below.

**Theorem 2.1** (Quantum Leverage Score Approximation, Apers & Gribling (2024, Theorem 3.2))**.** *Assume query access to a matrix $A = \mathbb{R}^{m \times n}$ with row sparsity $r$. For any $0 < \epsilon \leq 1$, there exists a quantum algorithm that, with high probability, provides quantum query access to estimates $\widetilde{\sigma}_i$ for any $i \in [m]$ satisfying $(1-\epsilon)\sigma_i(A) \leq \widetilde{\sigma}_i \leq (1+\epsilon)\sigma_i(A)$. The algorithm makes $\widetilde{O}(\sqrt{mn}/\epsilon)$ row queries to $A$ and runs in $\widetilde{O}(r\sqrt{mn}/\epsilon + n^\omega + \min\{n^\omega, nr^2\}/\epsilon^2 + n^2/\epsilon^4)$ time. Furthermore, the cost to estimate per $\widetilde{\sigma}_i$ is one row query to $A$ and $\widetilde{O}(r/\epsilon^2)$ time.*

We note that a row query to the $i$-th row of $A$ returns the entire row, and the query can be made in superposition. Equivalently, one row query corresponds to $O(r)$ queries to $\mathcal{O}_A$, since each row contains at most $r$ nonzero entries.

# 3. Quantum Algorithm for Multiscale Leverage Score Overestimates

In this section, we present our core quantum algorithm, *Quantum Multiscale Leverage Score Overestimates* (QMLSO), the key to enabling quantum acceleration in our GLM sparsification framework. We begin by defining MLSO, distilling the structural ideas from the approximate weight scheme of Jambulapati et al. (2024), which serves as the GLM analogue of importance scores. We then describe our quantum procedure for computing MLSO values from an initial coarse weight estimate, using quantum leverage score approximation to achieve sublinear-time refinement. Finally, we develop a quantum algorithm for constructing suitable initial weights, ensuring that the entire MLSO computation can be executed fully within the quantum setting.

## 3.1. Multiscale leverage score overestimates (MLSO)

We begin by recalling the definition of approximate weights introduced by Jambulapati et al. (2024), which generalizes the notion of $\ell_p$ Lewis weights to capture the "statistical contribution" of each row at a given scale, where the scale refers to the magnitude range of residuals being considered.

**Definition 3.1** (Approximate Weights, Jambulapati et al. (2024, Definition 1.7))**.** Fix a matrix $A \in \mathbb{R}^{m \times n}$ (where the $i$-th row is $a_i$ for each $i \in [m]$) and a family of loss functions $\mathcal{F} = \{f_1, \ldots, f_m\}$. We say a vector $w \in \mathbb{R}_+^m$ is an $\alpha$-approximate weight (with respect to $A$ and $\mathcal{F}$) at scale $s$ if, for all $i \in [m]$,

$$\frac{s}{\alpha} \leq \frac{f_i(\|M^{-1/2}a_i\|_2)}{w_i\|M^{-1/2}a_i\|_2^2} \leq \alpha s, \quad \text{where } M := \sum_{i=1}^{m} w_i a_i a_i^\top.$$

Since approximate weights are defined only at a single scale,

they are not sufficient for handling general loss functions where homogeneity fails. To address this, Jambulapati et al. (2024) introduced weight schemes—collections of approximate weights defined consistently across a range of scales. These schemes guarantee that the contribution of each row is well controlled not only at a single scale but simultaneously across multiple scales, thereby capturing the multiscale structure inherent in $A$ and $\mathcal{F}$.

**Definition 3.2** (Weight Schemes, Jambulapati et al. (2024, Definition 1.8))**.** Fix a matrix $A \in \mathbb{R}^{m \times n}$ (the $i$-th row is $a_i$ for each $i \in [m]$) and a family of loss functions $\mathcal{F} = \{f_1, \ldots, f_m\}$. Let $\mathcal{J} \subseteq \mathbb{Z}$ be a contiguous interval. We say that a family $\mathcal{W}_{A,\mathcal{F},\mathcal{J}} = \{w^{(j)} \in \mathbb{R}_+^m : j \in \mathcal{J}\}$ is an $\alpha$-approximate weight scheme if:

1. Each $w^{(j)}$ is an $\alpha$-approximate weight (with respect to $A$ and $f$) at scale $2^j$;

2. For every pair $j, j + 1 \in \mathcal{J}$, we have $w_i^{(j+1)} \leq \alpha w_i^{(j)}, \forall i \in [m]$.

We omit the subscript $(A, \mathcal{F}, \mathcal{J})$ when it is clear from context.

When working with weight schemes, we ultimately need a compact summary that upper bounds the statistical contribution of each row across all relevant scales. This motivates the notion of multiscale leverage score overestimates. By aggregating leverage scores over the entire weight scheme, such overestimates provide a single vector that controls row contributions across all scales. This will later serve as the key object enabling efficient sampling and sparsification in our quantum algorithm.

**Definition 3.3** (Multiscale Leverage Score Overestimates)**.** Let $A \in \mathbb{R}^{m \times n}$ be a matrix with rows $a_1, \ldots, a_m$, let $\mathcal{F} = \{f_1, \ldots, f_m\}$ be a family of loss functions, and let $\mathcal{J} \subseteq \mathbb{Z}$ be a contiguous interval. Suppose $\mathcal{W} = \{w^{(j)} \in \mathbb{R}_+^m : j \in \mathcal{J}\}$ is an $\alpha$-approximate weight scheme with respect to $A, \mathcal{F}, \mathcal{J}$. We say that a vector $z \in \mathbb{R}_+^m$ is a multiscale (leverage score) $\tau$-overestimates with respect to $A$ and $\mathcal{W}$ if

$$\|z\|_1 \leq \tau, \quad \text{and} \quad z_i \geq \sigma_i^{\text{ms}}(A, \mathcal{W}), \quad \forall i \in [m],$$

where the multiscale leverage score of the $i$-th row is defined as

$$\sigma_i^{\text{ms}}(A, \mathcal{W}) := \max_{j \in \mathcal{J}} \sigma_i(W_j^{1/2}A), \quad \forall i \in [m],$$

and $W_j := \text{diag}(w^{(j)})$ for each $j \in \mathcal{J}$.

## 3.2. Quantum algorithm for MLSO

We now present our quantum algorithm for computing multiscale leverage score overestimates (MLSO), which serves

as a key subroutine in our main quantum algorithm for GLM sparsification. Our approach builds upon the classical contractive algorithm of Cohen & Peng (2015); Jambulapati et al. (2024), but is carefully adapted to the quantum setting.

The input consists of quantum oracles for the matrix $A$ and the function family $\mathcal{F}$, along with query access to an initial approximate weight vector $w^\circ$ and relevant parameters. The output is a data structure that provides efficient quantum query access to multiscale leverage score overestimates (see Proposition C.3 for a formal description).

The algorithm proceeds iteratively: over $\widetilde{O}(1)$ rounds, it refines the largest-scale weight until it becomes well-conditioned. Once this condition is achieved, the algorithm recursively computes the weights at smaller scales and combines them to obtain the final MLSO. Within each iteration, the subroutine ModLevApprox supplies quantum query access to the modified leverage scores (Proposition C.1), while WeightCompute produces quantum query access to the updated weights for the subsequent round (Proposition C.2). In Algorithm 1, ModLevApprox is called only with the fixed internal accuracy $\eta = 0.1$, so the $\epsilon^{-4}$ dependence in Proposition C.1 contributes only to the non-sampling part of the QMLSO runtime. The complete procedure is summarized in Algorithm 1, and its formal guarantees are established in Theorem 3.4. Owing to space limitations, the proof of Theorem 3.4 is deferred to Appendix C.

---

**Algorithm 1** QMLSO($\mathcal{O}_\mathcal{F}, \mathcal{O}_A, \mathcal{O}_{w^\circ}, j_{\min}, j_{\max}, \beta, L, \theta, c$)

**Require:** A matrix oracle $\mathcal{O}_A$; a proper loss functions oracle $\mathcal{O}_\mathcal{F}$ with parameters $L, \theta, c$; an initial weight oracle $\mathcal{O}_{w^\circ}$ with associated parameters $\beta$ and $j_{\max}$; and an integer $j_{\min} < j_{\max}$.

**Ensure:** An instance $\mathcal{Z}$ of QOverestimate which stores the vector $z$ being a multiscale $O(n \cdot |j_{\max} - j_{\min}|)$-overestimate with respect to matrix $A$ and $\alpha$-approximate scheme $\mathcal{W} = \{w^{(j)} \in \mathbb{R}_+^m : j \in \mathcal{J} := \mathbb{Z} \cap [j_{\min}, j_{\max}]\}$ for some $\alpha = \alpha(L, \theta, c)$.

1: $\eta \leftarrow 0.1$, $\delta \leftarrow \max\left\{\frac{1}{2}, \left|\frac{\theta-2}{2}\right|\right\}$, $C \leftarrow \max\left\{\frac{2L}{c}, \frac{1}{c}\right\}$
2: $T \leftarrow \left\lceil \left(\log\log(\frac{1-\eta}{(1+\eta)^2} \cdot C) - \log\log\beta\right) / \log\delta \right\rceil$
3: $U_{w_{\mathrm{itr}}^{(0)}} \leftarrow \mathcal{O}_{w^\circ}$
4: **for** $i = 0$ to $T$ **do**
5: $\quad \mathcal{L}_{\mathrm{itr}}^{(i)} \leftarrow$ ModLevApprox($\mathcal{O}_A, U_{w_{\mathrm{itr}}^{(i)}}, \eta$)
6: $\quad U_{w_{\mathrm{itr}}^{(i+1)}} \leftarrow$ WeightCompute($\mathcal{O}_\mathcal{F}, U_{w_{\mathrm{itr}}^{(i)}}, \mathcal{L}_{\mathrm{itr}}^{(i)}, 2^{j_{\max}}$)
7: **end for**
8: $U_{w^{(j_{\max})}} \leftarrow U_{w_{\mathrm{itr}}^{(T)}}$
9: **for** $j = j_{\max} - 1, \ldots$ to $j_{\min}$ **do**
10: $\quad \mathcal{L}^{(j+1)} \leftarrow$ ModLevApprox($\mathcal{O}_A, U_{w^{(j+1)}}, \eta$)
11: $\quad U_{w^{(j)}} \leftarrow$ WeightCompute($\mathcal{O}_\mathcal{F}, U_{w^{(j+1)}}, \mathcal{L}^{(j+1)}, 2^j$)
12: **end for**
13: $\mathcal{Z} \leftarrow$ QOverestimate($\{\mathcal{L}^{(j)} : j \in \mathcal{J}\}, \eta$)

---

**Theorem 3.4** (Quantum Multiscale Leverage Score Overestimates). *There is a quantum algorithm* QMLSO($\mathcal{O}_\mathcal{F}, \mathcal{O}_A, \mathcal{O}_{w^\circ}, j_{\min}, j_{\max}, \beta, L, \theta, c$) *that, given query access $\mathcal{O}_A$ to a matrix $A = \mathbb{R}^{m \times n}$ (row sparsity $r \leq n$), query access $\mathcal{O}_\mathcal{F}$ to an $(L, \theta, c)$-proper loss functions family $\mathcal{F} = \{f_1, \ldots, f_m\}$, query access $\mathcal{O}_{w^\circ}$ to an initial weight $w^\circ \in \mathbb{R}_+^m$ which is a $\beta$-approximate weight at scale $2^{j_{\max}}$ (with respect to $A$ and $\mathcal{F}$) for some integer $j_{\max}$, and any integer $j_{\min} < j_{\max}$, the algorithm makes $O(\sqrt{mn})$ queries to $\mathcal{O}_{w^\circ}$, $O(r\sqrt{mn}\Delta)$ queries to $\mathcal{O}_A$, $O(\sqrt{mn}\Delta)$ queries to $\mathcal{O}_\mathcal{F}$, runs in time $\widetilde{O}\big((r\sqrt{mn} + n^\omega + nr^2)\Delta\big)$, where $\Delta = |j_{\max} - j_{\min}| + \max\{\log\log\beta, 0\}$. Then, w.h.p., it provides query access to a multiscale $O(n \cdot |j_{\max} - j_{\min}|)$-overestimates $z$ with respect to $A$ and $\alpha$-approximate weight scheme $\mathcal{W} = \{w^{(j)} \in \mathbb{R}_+^m : j \in \mathcal{J} := \mathbb{Z} \cap [j_{\min}, j_{\max}]\}$ for some $\alpha = \alpha(L, \theta, c)$, where each query to $z$ requires $\widetilde{O}(r |\mathcal{J}|)$ time.*

### 3.3. Initial Weight Construction

When applying our QMLSO algorithm, it is crucial to start with a well-conditioned $\beta$-approximate weight, where $\beta$ cannot be too large. In this subsection, we show how to construct such an initial weight by leveraging quantum leverage score approximation together with a binary search procedure, combined with a mild modification of the loss functions $f_i$. Our initialization algorithm outputs a quantum data structure that, given query access to $A$ and $\mathcal{F}$, as well as the largest scale $s_{\max}$ and a parameter $\delta$, provides quantum query access to an initial weight vector $w^\circ$ (at scale $s_{\max}$) together with the corresponding modified loss family $\mathcal{F}^\circ$. This construction can be viewed as the quantum analogue of the initialization procedure introduced by Jambulapati et al. (2024). We formalize this construction in the following theorem, and defer its proof to Appendix D.

**Theorem 3.5** (Weight Initialization). *Assume query access $\mathcal{O}_A$ to a matrix $A = \mathbb{R}^{m \times n}$ with row sparsity $r$ and query access $\mathcal{O}_\mathcal{F}$ to an $(L, \theta, c)$-proper loss functions family $\mathcal{F} = \{f_1, \ldots, f_m\}$. There exists a quantum data structure* WeightInitialize *that supports the following operations:*

1. *Initialization:* WeightInitialize($\mathcal{O}_A, \mathcal{O}_\mathcal{F}, s_{\max}, \delta$), *outputs an instance $\mathcal{A}$, which makes $\widetilde{O}(r\sqrt{mn})$ queries to $\mathcal{O}_A$ and $\widetilde{O}(\sqrt{mn})$ queries to $\mathcal{O}_\mathcal{F}$, in $\widetilde{O}(r\sqrt{mn} + n^\omega + nr^2)$ time.*

2. *Function query:* $\mathcal{A}$.QueryFunction, *provides query access $\mathcal{O}_{\mathcal{F}^\circ}$ to a $(\max\{1, L\}, \theta, c)$-proper loss functions*

family $\mathcal{F}^\circ = \{f_1^\circ, \ldots, f_m^\circ\}$ which satisfies

$$\sum_{i=1}^m f_i\left(\langle a_i, x\rangle\right) \le s_{\max}$$

$$\Rightarrow 0 \le \sum_{i=1}^m f_i^\circ\left(\langle a_i, x\rangle\right) - f_i\left(\langle a_i, x\rangle\right) \le C_{\text{INIT}}\delta m^2 s_{\max}, \tag{4}$$

for some constant $C_{\text{INIT}} = C_{init}(L, \theta, c)$. Each query requires $O(r)$ queries to $\mathcal{O}_A$ and $O(1)$ queries to $\mathcal{O}_{\mathcal{F}}$, and runs in $\widetilde{O}(r)$ time.

3. *Weight query:* $\mathcal{A}$.QueryWeight, *provides query access* $\mathcal{O}_{w^\circ}$ *to a weight* $w^\circ \in \mathbb{R}_+^m$ *which is a $\beta$-approximate weight at scale $s_{\max}$ with respect to $A$ and $\mathcal{F}^\circ$, where* $\beta = O((L/c)^2 m/\delta)$. *Each query requires $O(r)$ queries to $\mathcal{O}_A$ and $O(1)$ queries to $\mathcal{O}_{\mathcal{F}}$, and runs in* $\widetilde{O}(r)$ *time.*

## 4. Quantum Algorithm for Generalized Linear Models Sparsification

In this section, we present our *Quantum Algorithm for GLM Sparsification*, which builds on the MLSO procedure to achieve substantial quantum speedups. The first part outlines our quantum sparsification framework, which implements efficient importance sampling by combining the preparation of many copies of a quantum state (Hamoudi, 2022) with quantum sum estimation (Li et al., 2019), enabling sublinear-time sampling and reweighting. The second part demonstrates the versatility of this framework by embedding a broad class of optimization problems into it, showing that they can be solved with quantum speedups.

### 4.1. Quantum Algorithm for GLM Sparsification

To describe our quantum sparsification framework, we rely on two fundamental primitives. The first is Multiple Quantum State Preparation, which efficiently produces many copies of a quantum state; measuring these copies yields samples where each index $i$ is selected with probability proportional to $w_i$, thereby enabling importance sampling with sublinear overhead (restated in Theorem E.1). The second is Quantum Sum Estimation, which offers a quadratic speedup over classical approaches and is crucial for efficient reweighting in our framework (restated in Theorem E.2).

With these tools in place, we now present our main result: the Quantum Algorithm for GLM Sparsification. Building on the QMLSO procedure and the primitives above, our algorithm efficiently sparsifies generalized linear models while achieving polynomial quantum speedups. The algorithm and its formal guarantee are stated below, with the proof deferred to Appendix E; a compact overview of the full pipeline is given in Figure 1 of Appendix A.

---

**Algorithm 2** Quantum GLM Sparsification QGLMSparsify$(\mathcal{O}_\mathcal{F}, \mathcal{O}_A, L, \theta, c, \epsilon, s_{\min}, s_{\max})$

---

**Require:** A matrix oracle $\mathcal{O}_A$; a proper loss functions oracle $\mathcal{O}_\mathcal{F}$ with parameters $L, \theta, c$; parameters $\epsilon > 0, s_{\max} > s_{\min} \ge 0$.

**Ensure:** A non-negative weight vector $w \in \mathbb{R}_+^m$ such that $\widetilde{F}(x) := \sum_{i=1}^m w_i f_i(\langle a_i, x\rangle)$ is an $\widetilde{O}(\log(s_{\max}/s_{\min}) \cdot n/\epsilon^2)$-sparse $\epsilon$-approximate sparsifier of $F$ over the range $[s_{\min}, s_{\max}]$.

1: $\widetilde{w} \leftarrow 0, M \leftarrow \widetilde{\Theta}(n/\epsilon^2)$
2: $j_{\min} \leftarrow \lfloor \log s_{\min} - 4\log m \rfloor, j_{\max} \leftarrow \lceil \log s_{\max} \rceil$
3: $\delta \leftarrow O(\epsilon s_{\min}/(m^3 s_{\max})), \beta \leftarrow O((L/c)^2 m/\delta)$
4: $\mathcal{A} \leftarrow$ WeightInitialize$(\mathcal{O}_A, \mathcal{O}_\mathcal{F}, s_{\max}, \delta)$
   (Theorem 3.5)
5: $\mathcal{Z} \leftarrow$ QMLSO$(\mathcal{A}$.QueryFunction, $\mathcal{O}_A$,
   $\mathcal{A}$.QueryWeight, $j_{\min}, j_{\max}, \beta, L, \theta, c)$
   (Theorem 3.4)
6: $\vartheta \leftarrow$ MultiSample$(\mathcal{Z}, M)$   (Theorem E.1)
7: $\widetilde{\nu} \leftarrow$ SumEstimate$(\mathcal{Z}, 0.1)$   (Theorem E.2)
8: **for** $i = 1$ to $M$ **do**
9:    Get $z_{\vartheta_i}$ from $\mathcal{Z}$
10:    $\widetilde{w}_{\vartheta_i} \leftarrow \widetilde{w}_{\vartheta_i} + \widetilde{\nu}/(1.1\,M z_{\vartheta_i})$
11: **end for**
12: Return $\widetilde{w}$.

---

**Theorem 4.1** (Quantum Algorithm for GLM Sparsification). *Let $F(x) := \sum_{i=1}^m f_i\left(\langle a_i, x\rangle\right)$ be a total loss function defined by vectors $a_1, \ldots, a_m \in \mathbb{R}^n$ (with sparsity $r \le n$) and an $(L, \theta, c)$-proper loss functions family $\mathcal{F} = \{f_1, \ldots, f_m\}$. There exists a quantum algorithm* QGLMSparsify$(\mathcal{O}_A, \mathcal{O}_\mathcal{F}, L, \theta, c, \epsilon, s_{\min}, s_{\max})$ *that, given query access $\mathcal{O}_A$ to the matrix $A$ (where the $i$-row is $a_i$), query access $\mathcal{O}_\mathcal{F}$ to the function family $\mathcal{F}$ with parameters $L, \theta, c$, and numbers $\epsilon > 0$, $s_{\max} > s_{\min} \ge 0$, outputs w.h.p. a non-negative weight vector $w \in \mathbb{R}_+^m$ such that $\widetilde{F} := \sum_{i=1}^m w_i f_i(\langle a_i, x\rangle)$ is an $\widetilde{O}(\log(s_{\max}/s_{\min}) \cdot n/\epsilon^2)$-sparse $\epsilon$-approximate sparsifier of $F$ over the range $[s_{\min}, s_{\max}]$, in time $\widetilde{O}\left((n^\omega + nr^2 + r\sqrt{mn}/\epsilon) \cdot \log(s_{\max}/s_{\min})\right)$.*

We note that the $m$-dependent term improves over the classical $mr$ term when $\epsilon = \Omega(\sqrt{n/m})$, the regime where the sparsifier is smaller than the input; non-$m$ terms or access setup can dominate otherwise. The scale ratio term $\log(s_{\max}/s_{\min})$ can be removed when all $f_i$ are $p$-homogeneous for some $p > 0$ (i.e., $f_i(\lambda x) = |\lambda|^p f_i(x)$, and in particular $\ell_p$ is $p$-homogeneous). For non-homogeneous cases such as $\gamma_p$, this can be reduced to $\log(\text{poly}(m))$ via properness assumption (Jambulapati et al., 2024, Lemma 1.5).

*Remark* 4.2 (Reduction to Linear and $\ell_p$ Regression). In the special case of linear regression, the GLM sparsification framework only requires standard leverage-score overes-

timates rather than the multiscale leverage score overestimates (MLSO). Concretely, in Algorithm 2, Steps 2-5 can be replaced by the quantum leverage score routine of Apers & Gribling (2024); under this simplification, Algorithm 2 essentially reduces to a quantum spectral approximation procedure in the sense of Theorem 3.1 of Apers & Gribling (2024), and once a spectral sparsifier has been constructed, one can approximately solve linear regression and its variants (e.g., multiple and ridge regression) on top of that sparsifier using standard techniques. For the more general case of $\ell_p$ regression, the MLSO construction can likewise be reduced to Lewis weights, and Algorithm 1 can be viewed as a quantum iterative approach for producing approximate Lewis weights from approximate leverage scores; related quantum procedures for approximating Lewis weights have also been studied in Apers & Gribling (2024) and in the concurrent work on sublinear-time quantum sensitivity sampling of Song et al. (2025).

*Remark* 4.3 (Comparison with Quantum Lower Bounds). We briefly record two complementary quantum lower bounds that apply to our setting. Since linear regression is a special case of GLM sparsification (by taking $f_i = \ell_2$ for all $i$), any lower bound for quantum spectral approximation immediately yields a lower bound for quantum GLM sparsification. The first lower bound shows the optimality of the $m$-dependence for constant $\epsilon$: for arbitrary $m \geq n$, there exists a family of matrices $A \in \mathbb{R}^{m \times n}$ for which any quantum algorithm that computes a constant-factor approximation of $F$ must make $\Omega(\sqrt{mn})$ row queries to $A$ (Apers & Gribling, 2024, Theorem 8.1). The second lower bound shows optimality of the $\epsilon$-dependence, but it is restricted to $m \in O(n^2)$: for arbitrary $n$ and $m \in O(n^2)$ and any $\epsilon \in \Omega(\sqrt{m/n})$, there exists a family of matrices $A \in \mathbb{R}^{m \times n}$ for which any quantum algorithm that computes an $\epsilon$-approximation of $F$ must make $\Omega(\sqrt{mn}/\epsilon)$ row queries to $A$ (Apers & de Wolf, 2022, Theorem 2).

## 4.2. Applications

In this section, we demonstrate how our quantum GLM framework accelerates a wide range of regression tasks. We begin with the general $\ell_p$ and $\gamma_p$ loss functions, and then specialize to linear regression and its variants, including multiple, ridge, and Lasso regression.

**General $\ell_p$ and $\gamma_p$ Regression.** We first show that the GLM framework captures the broad class of $\ell_p$ and $\gamma_p$ regressions. Observe that both $\ell_p(x) = |x|^p$ and $\gamma_p$ (defined in Equation (2)) are $(1, p/2, 1)$-proper for $p \in (0, 2]$. Consequently, Theorem 4.1 yields an $\widetilde{O}(n/\epsilon^2)$-sparse $\epsilon$-approximate sparsifier, computable in time $\widetilde{O}(r\sqrt{mn}/\epsilon + n^\omega + nr^2)$. The problem then reduces to a small instance solvable classically in $\text{poly}(n, 1/\epsilon)$ time. This yields the following quantum speedups:

**Corollary 4.4** (Quantum $\ell_p$ Regression). *Let $p \in (0, 2]$. There exists a quantum algorithm that, given query access to $A \in \mathbb{R}^{m \times n}$ (row sparsity $r$) and $b \in \mathbb{R}^m$, outputs w.h.p. an $\epsilon$-approximate solution $x \in \mathbb{R}^n$ minimizing $\|Ax - b\|_p^p$, in $\widetilde{O}(r\sqrt{mn}/\epsilon) + \text{poly}(n, \epsilon)$ time.*

**Corollary 4.5** (Quantum $\gamma_p$ Regression). *Let $p \in (0, 2]$. There exists a quantum algorithm that, given query access to $A \in \mathbb{R}^{m \times n}$ (row sparsity $r$) and $b \in \mathbb{R}^m$, outputs w.h.p. an $\epsilon$-approximate solution $x \in \mathbb{R}^n$ minimizing $F(x) = \sum_{i=1}^m \gamma_p(\langle a_i, x \rangle - b_i)$, in $\widetilde{O}(r\sqrt{mn}/\epsilon) + \text{poly}(n, \epsilon)$ time. Notably, the case $p = 1$ corresponds to Huber regression.*

**Linear Regression** ($p = 2$). As discussed in Remark 4.2, for $p = 2$ our framework simplifies to standard quantum spectral sparsification. Combining this with a fast classical solver (Clarkson & Woodruff, 2017, Theorem 43), while noting that $m' < m$ implies $\epsilon = \Omega(\sqrt{n/m})$, yields:

**Corollary 4.6** (Quantum Linear Regression). *There exists a quantum algorithm that, given query access to $A \in \mathbb{R}^{m \times n}$ (row sparsity $r$) and $b \in \mathbb{R}^m$, outputs w.h.p. an $\epsilon$-approximate solution $x \in \mathbb{R}^n$ minimizing $\|Ax - b\|_2$, in $\widetilde{O}(r\sqrt{mn}/\epsilon + n^3)$ time.*

**Multiple Regression.** We next consider multiple regression, where the goal is to find $X \in \mathbb{R}^{n \times N}$ minimizing $\|AX - B\|_F$. Writing $X = [x^1, \ldots, x^N]$ and $B = [b^1, \ldots, b^N]$, we expand the squared Frobenius norm as $\|AX - B\|_F^2 = \sum_{k=1}^N \|Ax^k - b^k\|^2 = \sum_{k=1}^N \sum_{j=1}^m (\langle a_i, x^k \rangle - B_{jk})^2$. This reduces to a linear regression with sample size $m' = mN$; however, since sampling probabilities depend solely on $A$, we generate a single sparsifier for all $N$ targets. This avoids an $N$-factor quantum overhead before invoking a classical solver (e.g., (Clarkson & Woodruff, 2017, Theorem 38)).

**Corollary 4.7** (Quantum Multiple Regression). *There exists a quantum algorithm that, given query access to $A \in \mathbb{R}^{m \times n}$ (row sparsity $r$) and $B \in \mathbb{R}^{m \times N}$, outputs w.h.p. an $\epsilon$-approximate solution $X \in \mathbb{R}^{n \times N}$ minimizing $\|AX - B\|_F$, in $\widetilde{O}\left(r\sqrt{mn}/\epsilon + n^2(Nn + n/\epsilon + N/\epsilon)\right)$ time.*

**Ridge Regression.** The ridge regression objective is $F_{\text{ridge}}(x) = \|Ax - b\|_2^2 + \lambda\|x\|_2^2$. The regularization term can be interpreted as appending virtual data points. We define the augmented matrices:

$$A' = \begin{bmatrix} A \\ \sqrt{\lambda} I_n \end{bmatrix} \in \mathbb{R}^{(m+n) \times n}, \qquad b' = \begin{bmatrix} b \\ 0_n \end{bmatrix} \in \mathbb{R}^{m+n}. \quad (5)$$

By choosing the quadratic loss $f(x) = x^2$, we have $\sum_{i=1}^{m+n} f(\langle a_i', x \rangle - b_i') = F_{\text{ridge}}(x)$. Thus, ridge regression reduces to linear regression with effective sample size $m' = m + n$.

**Corollary 4.8** (Quantum Ridge Regression). *Let $\lambda > 0$. There exists a quantum algorithm that, given query access*

*to $A \in \mathbb{R}^{m \times n}$ (row sparsity $r$) and $b \in \mathbb{R}^m$, outputs w.h.p. an $\epsilon$-approximate solution $x \in \mathbb{R}^n$ minimizing $\|Ax - b\|_2^2 + \lambda\|x\|_2^2$, in $\widetilde{O}(r\sqrt{mn}/\epsilon + n^3)$ time.*

**Lasso Regression.** Finally, for Lasso regression with objective $F_{\mathrm{lasso}}(x) = |Ax - b|_2^2 + \lambda|x|_1$, we adopt the same augmented notation $A'$, $b'$ from Equation (5) but assign different loss functions: quadratic loss $f_i'(z) = z^2$ for the first $m$ rows, and $\ell_1$ loss $f_i'(z) = |z|_1$ for the remaining $n$ rows. This fits directly into our GLM framework.

**Corollary 4.9** (Quantum Lasso Regression)**.** *Let $\lambda > 0$. There exists a quantum algorithm that, given query access to $A \in \mathbb{R}^{m \times n}$ (row sparsity $r$) and $b \in \mathbb{R}^m$, outputs w.h.p. an $\epsilon$-approximate solution $x \in \mathbb{R}^n$ minimizing $\|Ax - b\|_2^2 + \lambda\|x\|_1$, in $\widetilde{O}(r\sqrt{mn}/\epsilon + n^3/\epsilon^2)$ time.*

## Impact Statement

This paper presents work whose goal is to advance the field of Machine Learning. There are many potential societal consequences of our work, none which we feel must be specifically highlighted here.

## Acknowledgements

The work is supported by National Key Research and Development Program of China (Grant No. 2023YFA1009403), National Natural Science Foundation of China (Grant No. 12347104), and Beijing Science and Technology Planning Project (Grant No. Z25110100810000).

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

# A. Algorithmic pipeline

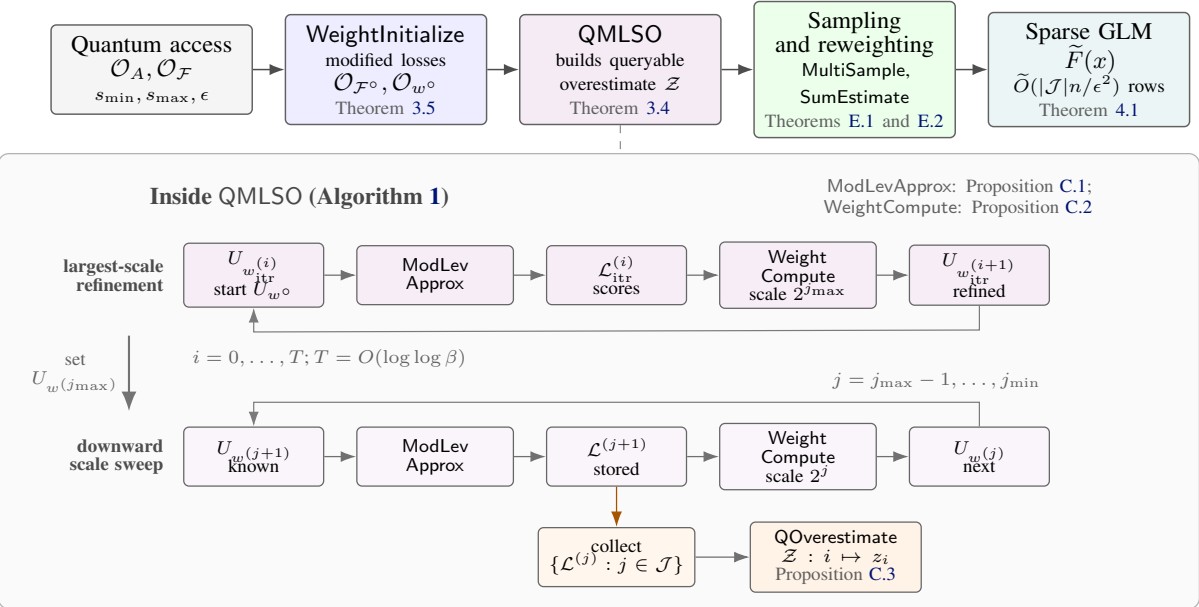

*Figure 1.* A schematic view of the quantum GLM sparsification pipeline. The top row shows the main flow of Algorithm 2. The enlarged QMLSO panel highlights the two update mechanisms in Algorithm 1: a contractive iteration at the largest scale, followed by a downward recursion across scales. Each update uses ModLevApprox and WeightCompute, and the resulting leverage-score data structures are collected into the queryable overestimate $\mathcal{Z}$.

# B. Quantum computing

In quantum mechanics, a $d$-dimensional quantum state $|v\rangle = (v_0, \ldots, v_{d-1})^\top$ is represented as a unit vector in the Hilbert space $\mathbb{C}^d$, satisfying $\sum_{i \in [d]_0} |v_i|^2 = 1$. The computational basis of this space is $\{|i\rangle\}_{i \in [d]_0}$, where $|i\rangle = (0, \ldots, 0, 1, 0, \ldots, 0)^\top$ with the $i$-th coordinate (0-indexed) equal to 1 and all others 0. For two states $|u\rangle, |v\rangle \in \mathbb{C}^d$, their inner product is $\langle u|v\rangle = \sum_{i \in [d]_0} u_i^* v_i$, with $z^*$ denoting complex conjugation. The tensor product of $|u\rangle \in \mathbb{C}^{d_1}$ and $|v\rangle \in \mathbb{C}^{d_2}$ is their Kronecker product, $|u\rangle \otimes |v\rangle = (u_0 v_0, u_0 v_1, \ldots, u_{d_1-1} v_{d_2-1})^\top$, often abbreviated as $|u\rangle|v\rangle$.

A quantum bit, or qubit, is simply a quantum state in $\mathbb{C}^2$, expressible as $|\psi\rangle = \alpha|0\rangle + \beta|1\rangle$, where $\alpha, \beta \in \mathbb{C}$ and $|\alpha|^2 + |\beta|^2 = 1$. An $n$-qubit state belongs to the tensor product space $(\mathbb{C}^2)^{\otimes n} = \mathbb{C}^{2^n}$, with basis states $|i\rangle_{i \in [2^n]_0}$. When measuring an $n$-qubit state $|\psi\rangle$ in this basis, the outcome $i$ is observed with probability $|\langle i|\psi\rangle|^2$. Quantum operations are described by unitary operators $U$, which satisfy $UU^\dagger = U^\dagger U = I$, where $U^\dagger$ denotes the conjugate transpose and $I$ is the identity operator.

In many quantum algorithms, information is stored and accessed through quantum-read classical-write random access memory (QRAM) (Giovannetti et al., 2008), which is employed in numerous quantum algorithms. QRAM allows the storage and modification of a classical array $a_1, \ldots, a_n$ while supporting quantum query access via the unitary $U_{\text{QRAM}}$ : $|i\rangle|0\rangle \mapsto |i\rangle|a_i\rangle$. If one is only concerned with quantum query complexity, the QRAM assumption can be removed at the cost of a polynomial increase in time complexity. While QRAM serves as a natural quantum analogue of the classical RAM model and is a standard assumption in theoretical quantum algorithms, constructing scalable QRAM remains a significant engineering challenge. Recent experimental works explore QRAM-like architectures, including transmon-controlled phonon routing and bucket-brigade schemes on superconducting processors (Wang et al., 2025; Shen et al., 2025; Zhang et al., 2025); nevertheless, we use QRAM here as a theoretical access model rather than a near-term practical assumption.

A quantum (query) algorithm $\mathcal{A}$ is a quantum circuit consisting of a sequence of unitaries $U_1, \ldots, U_T$, where each $U_t$ may represent a quantum gate, a quantum oracle, or a QRAM operation. The time complexity of $\mathcal{A}$ is measured by the total number $T$ of such operations. The algorithm acts on $n$ qubits, initialized in the state $|0\rangle^{\otimes n}$. Applying the unitaries sequentially produces the final state $|\psi\rangle = U_T \ldots U_1 |0\rangle^{\otimes n}$. Finally, a measurement in the computational basis $|i\rangle_{i \in [2^n]_0}$

then yields a classical outcome $i$ with probability $|\langle i|\psi\rangle|^2$.

## C. Proof of Theorem 3.4

### C.1. Quantum Subroutines

To present our algorithm more clearly, we introduce the following data structure, which can be viewed as a straightforward application of Theorem 2.1.

**Proposition C.1** (Quantum Modified Leverage Score Approximation). *Assume query access $\mathcal{O}_A$ to a matrix $A = \mathbb{R}^{m \times n}$ with row sparsity $r$ and query access $\mathcal{O}_w$ to a vector $w \in \mathbb{R}_+^m$. For any $0 < \epsilon \leq 1$, there exists a quantum data structure* ModLevApprox *that supports the the following operations:*

1. *Initialization:* ModLevApprox$(\mathcal{O}_A, \mathcal{O}_w, \epsilon)$, *outputs an instance $\mathcal{L}$, making $\widetilde{O}(r\sqrt{mn}/\epsilon)$ queries to $\mathcal{O}_A$ and $\widetilde{O}(\sqrt{mn}/\epsilon)$ queries to $\mathcal{O}_w$, in $\widetilde{O}(r\sqrt{mn}/\epsilon + (n^\omega + nr^2)/\epsilon^2 + n^2/\epsilon^4)$ time.*

2. *Query:* $\mathcal{L}$.Query, *outputs a unitary satisfying*

$$\mathcal{L}.\text{Query}|i\rangle|0\rangle = |i\rangle|\widetilde{\sigma}_i\rangle,$$

*where for all $i \in [m]$, it holds that $(1 - \epsilon)\sigma_i \leq \widetilde{\sigma}_i \leq (1 + \epsilon)\sigma_i$, with $\sigma_i = \sigma_i(W^{1/2}A)$ denoting the leverage score, and $W = \text{diag}(w)$. Each query requires $O(r)$ queries to $\mathcal{O}_A$ and $O(1)$ queries to $\mathcal{O}_w$, and runs in $\widetilde{O}(r/\epsilon^2)$ time.*

*Proof of Proposition C.1.* Write $B := W^{1/2}A \in \mathbb{R}^{m \times n}$, where $W = \text{diag}(w)$. Because $W^{1/2}$ rescales each row of $A$ by $\sqrt{w_i}$, $B$ has the *same* row sparsity $r$. Let $A_i$ be the $i$-th row of $A$. Fix a row index $i$ and let $S_i = \text{supp}(A_i.)$ with $|S_i| = r_i \leq r$. All entries of $A_i$ can be obtained using $O(r)$ queries to $\mathcal{O}_A$, one query to $\mathcal{O}_w$, and $\widetilde{O}(r)$ arithmetic time. For the superposition, we can perform

$$|i\rangle|0\rangle \xmapsto{\mathcal{O}_A^{\text{idx}}} \frac{1}{\sqrt{r_i}} \sum_{j \in S_i} |i\rangle|j\rangle|0\rangle$$

$$\xmapsto{\mathcal{O}_A^{\text{elem}}} \frac{1}{\sqrt{r_i}} \sum_{j \in S_i} |i\rangle|j\rangle|A_{ij}\rangle|0\rangle$$

$$\xmapsto{\mathcal{O}_w} \frac{1}{\sqrt{r_i}} \sum_{j \in S_i} |i\rangle|j\rangle|A_{ij}\rangle|w_i\rangle|0\rangle$$

$$\xmapsto{U_{\text{MULT}}} \frac{1}{\sqrt{r_i}} \sum_{j \in S_i} |i\rangle|j\rangle|A_{ij}\rangle|w_i\rangle|\sqrt{w_i}A_{ij}\rangle$$

$$\xmapsto{\mathcal{O}_w^\dagger, \mathcal{O}_A^{\text{elem}\,\dagger}, \mathcal{O}_A^{\text{idx}\,\dagger}} \frac{1}{\sqrt{r_i}} \sum_{j \in S_i} |i\rangle|j\rangle|0\rangle|0\rangle|B_{ij}\rangle.$$

Thus one row query in Theorem 2.1 corresponds to $O(r)$ queries to $\mathcal{O}_A$, $O(1)$ queries to $\mathcal{O}_w$, plus $\widetilde{O}(r)$ arithmetic time. Therefore, invoking the Theorem 2.1 on $B$ with accuracy $\epsilon$ gives, with high probability: (i) an initialisation that performs $\widetilde{O}(\sqrt{mn}/\epsilon)$ row queries to $B$ and runs in $\widetilde{O}(r\sqrt{mn}/\epsilon + (n^\omega + nr^2)/\epsilon^2 + n^2/\epsilon^4)$; substituting the row-query cost yields $\widetilde{O}(r\sqrt{mn}/\epsilon)$ calls to $\mathcal{O}_A$ and $\widetilde{O}(\sqrt{mn}/\epsilon)$ calls to $\mathcal{O}_w$; (ii) a query unitary that maps $|i\rangle|0\rangle$ to $|i\rangle|\widetilde{\sigma}_i(B)\rangle$ with $(1 - \epsilon)\sigma_i(B) \leq \widetilde{\sigma}_i(B) \leq (1 + \epsilon)\sigma_i(B)$, using one row query to $B$. Translating again, each query costs $O(r)$ calls to $\mathcal{O}_A$, $O(1)$ call to $\mathcal{O}_w$, and $\widetilde{O}(r/\epsilon^2)$ time. $\qquad\square$

The following proposition describes the weight update step in the algorithm QMLSO (Algorithm 1).

**Proposition C.2** (Weight Computation for Overestimates). *Let $s > 0$. Assume query access $\mathcal{O}_\mathcal{F}$ to the function family $\mathcal{F} = \{f_1, \ldots, f_m\}$ and query access $\mathcal{O}_w$ to a vector $w \in \mathbb{R}_+^m$. Let $\mathcal{L}$ denote an instance of* ModLevApprox. *Then, there exists a quantum algorithm* WeightCompute$(\mathcal{O}_\mathcal{F}, \mathcal{O}_w, \mathcal{L}, s)$ *such that*

$$\text{WeightCompute}(\mathcal{O}_\mathcal{F}, \mathcal{O}_w, \mathcal{L}, s)|i\rangle|0\rangle = |i\rangle|w_i'\rangle$$

*where*

$$w_i' = \frac{1}{s} \cdot \frac{f_i(\sqrt{\sigma_i/w_i})}{\sigma_i/w_i},$$

*and $\sigma_i$ is the leverage score returned by querying $\mathcal{L}$ at index $i \in [m]$. The algorithm makes $O(1)$ queries to $\mathcal{L}$, $\mathcal{O}_w$, and $\mathcal{O}_{\mathcal{F}}$, and requires an additional $\widetilde{O}(1)$ time for computation.*

*Proof of Proposition C.2.* Consider the following procedure:

$$
\begin{aligned}
|i\rangle|0\rangle &\xmapsto{\mathcal{L}.\mathsf{Query}} |i\rangle|\sigma_i\rangle|0\rangle \\
&\xmapsto{\mathcal{O}_w} |i\rangle|\sigma_i\rangle|w_i\rangle|0\rangle \\
&\xmapsto{U_{\mathrm{DIV}}} |i\rangle|\sigma_i\rangle|w_i\rangle|\sigma_i/w_i\rangle|0\rangle|0\rangle \\
&\xmapsto{U_{\mathrm{SQRT}}} |i\rangle|\sigma_i\rangle|w_i\rangle|\sigma_i/w_i\rangle|\sqrt{\sigma_i/w_i}\,\rangle|0\rangle \\
&\xmapsto{\mathcal{O}_{\mathcal{F}}} |i\rangle|\sigma_i\rangle|w_i\rangle|\sigma_i/w_i\rangle|\sqrt{\sigma_i/w_i}\,\rangle|f_i(\sqrt{\sigma_i/w_i}\,)\rangle \\
&\xmapsto{U_{\mathrm{DIV}(sr)}} |i\rangle|0\rangle|0\rangle|0\rangle|0\rangle|w_i'\rangle,
\end{aligned}
$$

where $U_{\mathrm{DIV}}$ computes the division $\sigma_i/w_i$, $U_{\mathrm{SQRT}}$ computes square root, and $U_{\mathrm{DIV}(sr)}$ carries out the arithmetic $w_i' := \frac{1}{s} \cdot \frac{f_i(\sqrt{\sigma_i/w_i}\,)}{r}$ and then uncomputes the ancillary registers $\sigma_i, w_i, \sigma_i/w_i, \sqrt{\sigma_i/w_i}$.

The computation uses $O(1)$ queries to $\mathcal{L}$, $\mathcal{O}_w$, and $\mathcal{O}_{\mathcal{F}}$, and $\widetilde{O}(1)$ additional arithmetic operations, as claimed in the proposition. □

The following proposition specifies the output data structure produced by QMLSO (Algorithm 1).

**Proposition C.3** (Preparation for Multiscale Overestimates). *Let $0 < \epsilon < 1$, $\mathcal{J} = \{j_{\min}, \dots, j_{\max}\} \subseteq \mathbb{Z}$ be a contiguous interval. Suppose we are given a collection of instances $\{\mathcal{L}^{(j)} : j \in \mathcal{J}\}$, where each $\mathcal{L}^{(j)}$ is an instance of $\mathsf{ModLevApprox}$. Then, there exists a quantum data structure $\mathsf{QOverestimate}$ that supports the following operations:*

- *Initialization: $\mathsf{QOverestimate}(\{\mathcal{L}^{(j)} : j \in \mathcal{J}\}, \epsilon)$, outputs an instance $\mathcal{Z}$ in total time $\widetilde{O}(\sum_{j \in \mathcal{J}} \zeta_j)$, where $\zeta_j$ denotes the needed time of initialization of $\mathcal{L}^{(j)}$.*

- *Query: $\mathcal{Z}.\mathsf{Query}$, outputs a unitary such that for every $i \in [m]$,*

$$\mathcal{Z}.\mathsf{Query}|i\rangle|0\rangle = |i\rangle|z_i\rangle,$$

  *where*

$$z_i = \frac{1}{1-\epsilon} \sum_{j \in \mathcal{J}} \sigma_i^{(j)},$$

  *and $\sigma_i^{(j)}$ is leverage score returned by querying $\mathcal{L}^{(j)}$ at index $i \in [m]$. This query is executed in $\widetilde{O}(\sum_{j \in \mathcal{J}} \iota_j)$ time, where $\iota_j$ represents the time required to querying $\mathcal{L}^{(j)}$.*

*Proof of Proposition C.3.* For every $j \in \mathcal{J}$, suppose we hold an instance $\mathcal{L}^{(j)}$ of $\mathsf{ModLevApprox}$. Denote by $\zeta_j$ its initialization cost and by $\iota_j$ the cost of a single query. The initialization of $\mathsf{QOverestimate}$ consists of initializing all instances $\mathcal{L}^{(j)}$ for $j \in \mathcal{J}$ and storing $\epsilon$ in QRAM in $\widetilde{O}(1)$ time, allowing access through a unitary $U_\epsilon : |i\rangle|0\rangle \to |i\rangle|1/1-\epsilon\rangle$.

For the query operation, on input $|i\rangle|0\rangle$, attach an accumulator register initialized to $|0\rangle$, and iterate over the scales $j \in \mathcal{J}$:

$$|i\rangle|0\rangle \xmapsto{\mathcal{L}^{(j_{\min})}.\text{Query}} |i\rangle|\sigma_i^{(j_{\min})}\rangle|0\rangle \xmapsto{U_{\text{ADD}}} |i\rangle|\sigma_i^{(j_{\min})}\rangle|\sigma_i^{(j_{\min})}\rangle|0\rangle$$

$$\xmapsto{(\mathcal{L}^{(j_{\min})}.\text{Query})^\dagger} |i\rangle|0\rangle|\sigma_i^{(j_{\min})}\rangle|0\rangle$$

$$\xmapsto{\mathcal{L}^{(j_{\min}+1)}.\text{Query}} |i\rangle|\sigma_i^{(j_{\min}+1)}\rangle|\sigma^{(j_{\min})}\rangle|0\rangle \xmapsto{U_{\text{ADD}}} |i\rangle|\sigma_i^{(j_{\min}+1)}\rangle|\sigma^{(j_{\min})} + \sigma_i^{(j_{\min}+1)}\rangle|0\rangle$$

$$\xmapsto{(\mathcal{L}^{(j_{\min}+1)}.\text{Query})^\dagger} |i\rangle|0\rangle|\sigma_i^{(j_{\min})} + \sigma_i^{(j_{\min}+1)}\rangle|0\rangle$$

$$\xmapsto{\cdots} |i\rangle|0\rangle|\sum_{j \in \mathcal{J}} \sigma_i^{(j)}\rangle|0\rangle$$

$$\xmapsto{U_\epsilon} |i\rangle|1/(1-\epsilon)\rangle|\sum_{j \in \mathcal{J}} \sigma_i^{(j)}\rangle|0\rangle \xmapsto{U_{\text{MULT}}} |i\rangle|1/(1-\epsilon)\rangle|\sum_{j \in \mathcal{J}} \sigma_i^{(j)}\rangle|z_i\rangle,$$

The adder $U_{\text{ADD}}$ and multiplier $U_{\text{MULT}}$ are implemented using fixed-point arithmetic and costs $\widetilde{O}(1)$ gates. The query procedure calls each $\mathcal{L}^{(j)}$ once and invokes its inverse once; together this costs $\Theta(\iota_j)$. Summing over all $j \in \mathcal{J}$, the total runtime is $O(\sum_{j \in \mathcal{J}} \iota_j)$. The arithmetic overhead from the adders contributes $\widetilde{O}(|\mathcal{J}|)$ additional gates, which is absorbed in the stated complexity. $\qquad\square$

## C.2. Proof of Theorem 3.4

We first analyze the time complexity. Since $\theta \in (0, 4)$, we have $\delta = \max\left\{\frac{1}{2}, \left|\frac{\theta-2}{2}\right|\right\} < 1$, and thus $T = \Theta_{L,\theta,c}(\log\log\beta)$. In each iteration $i \leq T$, the subroutine ModLevApprox runs in $\widetilde{O}(r\sqrt{mn} + n^\omega + nr^2)$ time (absorbing the cost of queries), and provides query access such that each query requires $O(r)$ calls to $\mathcal{O}_A$, $O(1)$ calls to $\mathcal{O}_{w_{\text{itr}}^{(i)}}$, and $\widetilde{O}(r)$ time (Proposition C.1). The subroutine WeightCompute then constructs $U_{w_{\text{itr}}^{(i+1)}}$, where each query requires $\widetilde{O}(r)$ time. This is because it makes $O(1)$ calls to $\mathcal{L}_{\text{itr}}^{(i)}$ and $\mathcal{O}_{w_{\text{itr}}^{(i)}}$, each of which can be executed in $\widetilde{O}(r)$ time (Proposition C.2). For the downward recursion across $|\mathcal{J}|$ scales (lines 9-12 of Algorithm 1), the complexity is the same per step. Therefore, the total cost of iterations and recursion is $\widetilde{O}\left((r\sqrt{mn} + n^\omega + nr^2)\Delta\right)$, making $O(\sqrt{mn})$ queries to $\mathcal{O}_{w^\circ}$, $O(r\sqrt{mn}\Delta)$ queries to $\mathcal{O}_A$, $O(\sqrt{mn}\Delta)$ queries to $\mathcal{O}_\mathcal{F}$, where $\Delta = |j_{\max} - j_{\min}| + \max\{\log\log\beta, 0\}$. In the final step of the algorithm, the data structure $\mathcal{Z}$ can be initialized in $\widetilde{O}\left((r\sqrt{mn} + n^\omega + nr^2)|\mathcal{J}|\right)$ time, and each call to $\mathcal{Z}$.Query can be executed in $\widetilde{O}(r|\mathcal{J}|)$ time. This follows directly from Proposition C.3, with parameters $\zeta_t = \widetilde{O}(r\sqrt{mn} + n^\omega + nr^2)$ and $\iota_t = \widetilde{O}(r)$.

In the remainder of this section, we will show that the vector $z \in \mathbb{R}^m$ corresponding to the output of QMLSO (Algorithm 1) is indeed an $O(n \cdot |j_{\max} - j_{\min}|)$-overestimate with respect to $A$ and an $\alpha$-approximate scheme $\mathcal{W}$. To analyze the correctness of the algorithm, we first introduce the following metric.

**Definition C.4.** Define a metric $d$ on $\mathbb{R}_+^m$ by

$$d(u, w) := \max\left\{\left|\log\frac{u_i}{w_i}\right| : i \in [m]\right\}$$

It is straightforward to verify that $d$ is indeed a metric, as it satisfies symmetry, positivity, and the triangle inequality.

A useful characterization is the following, which is immediate from the definition: it shows that being an $\alpha$-approximate weight is equivalent to requiring that $w$ is almost a fixed point of the update map $\phi_s$, up to distance $\log\alpha$ in the metric $d$.

*Fact* 1. Fix a matrix $A \in \mathbb{R}^{m \times n}$ (where the $i$-th row is $a_i$ for each $i \in [m]$) and a family of loss functions $\mathcal{F} = \{f_1, \ldots, f_m\}$. A vector $w \in \mathbb{R}_+^m$ is an $\alpha$-approximate weight (with respect to $A$ and $\mathcal{F}$) at scale $s$ if and only if

$$d(w, \phi_s(w)) \leq \log\alpha$$

where $\phi_s : \mathbb{R}_+^m \to \mathbb{R}_+^m$ is the update function defined by

$$(\phi_s(w))_i := \frac{1}{s} \cdot \frac{f_i(\sqrt{\sigma_i(W^{1/2}A)/w_i})}{\sigma_i(W^{1/2}A)/w_i}, \quad \text{and} \quad W := \text{diag}(w).$$

Furthermore, $\mathcal{W} = \{w^{(j)} \in \mathbb{R}_+^m : j \in \mathcal{J}\}$ is an $\alpha$-approximate weight scheme if and only if

$$\forall j \in \mathcal{J}: \ d\big(w^{(j)}, \phi_{2^j}\big(w^{(j)}\big)\big) \leq \log \alpha; \tag{6}$$

$$\forall j, j+1 \in \mathcal{J}: \ d\big(w^{(j+1)}, w^{(j)}\big) \leq \log \alpha. \tag{7}$$

In the noisy setting (where Algorithm 1 actually operates), we define the update function $\widetilde{\phi}_{s,\epsilon} : \mathbb{R}_+^m \to \mathbb{R}_+^m$ by

$$\left(\widetilde{\phi}_{s,\epsilon}(w)\right)_i := \frac{1}{s} \frac{f_i\left(\sqrt{\widetilde{\sigma}_i/w_i}\right)}{\widetilde{\sigma}_i/w_i}, \quad \forall i \in [m], \tag{8}$$

where $\widetilde{\sigma}_i$ is an $\epsilon$-approximation of $\sigma_i(W^{1/2}A)$, i.e., $(1-\epsilon)\sigma_i(W^{1/2}A) \leq \widetilde{\sigma}_i \leq (1+\epsilon)\sigma_i(W^{1/2}A)$.

Our analysis builds on the ideas of Jambulapati et al. (2024). We use the following result, which shows that the weight update procedure remains "contractive" even when noise is present.

**Lemma C.5** (Contraction of a noisy update, Lemma 2.9 of Jambulapati et al. (2024)). *Let* $\delta = \max\left\{\frac{1}{2}, \left|\frac{\theta-2}{2}\right|\right\}$, $C = \max\left\{\frac{2L}{c}, \frac{1}{c}\right\}$. *For any* $w, w' \in \mathbb{R}_+^m$ *and* $\epsilon, \epsilon' \in [0,1)$, *it holds that* $d(\widetilde{\phi}_{s,\epsilon}(w), \widetilde{\phi}_{s,\epsilon'}(w')) \leq \delta \cdot d(w, w') + \log C + \log \frac{1+\epsilon}{1-\epsilon'}$.

In the following application to Algorithm 1, the noisy updates use the fixed constant $\eta = 0.1$ from line 1 of the algorithm. At the beginning, assuming the initial weight is $\beta$-approximate, we apply the triangle inequality of $d$ together with Lemma C.5 to get

$$d\left(w_{\text{itr}}^{(0)}, \widetilde{\phi}_{s_{\max},\eta}\left(w_{\text{itr}}^{(0)}\right)\right) \leq d\left(w_{\text{itr}}^{(0)}, \phi_{s_{\max}}\left(w_{\text{itr}}^{(0)}\right)\right) + d\left(\phi_{s_{\max}}\left(w_{\text{itr}}^{(0)}\right), \widetilde{\phi}_{s_{\max},\eta}\left(w_{\text{itr}}^{(0)}\right)\right)$$
$$\leq \log \beta + \log C + \log(1+\eta).$$

For the iterative procedure (lines 4-7 of Algorithm 1), by applying Lemma C.5 inductively, we obtain

$$d\left(w_{\text{itr}}^{(T)}, \widetilde{\phi}_{s_{\max},\eta}\left(w_{\text{itr}}^{(T)}\right)\right) \leq \delta^T \cdot d\left(w_{\text{itr}}^{(0)}, \widetilde{\phi}_{s_{\max},\eta}\left(w_{\text{itr}}^{(0)}\right)\right) + \sum_{i=0}^{T-1} \delta^i \left(\log C + \log \frac{1+\eta}{1-\eta}\right)$$
$$\leq \delta^T \cdot \log \beta + \sum_{i=0}^{T} \delta^i \left(\log C + \log \frac{1+\eta}{1-\eta}\right)$$
$$\leq \frac{\delta^T \log \beta + \log C + \log \frac{1+\eta}{1-\eta}}{1-\delta}.$$

Therefore, after $T$ iterations, we have

$$d\left(w_{\text{itr}}^{(T)}, \phi_{s_{\max}}\left(w_{\text{itr}}^{(T)}\right)\right) \leq d\left(w_{\text{itr}}^{(T)}, \widetilde{\phi}_{s_{\max},\eta}\left(w_{\text{itr}}^{(T)}\right)\right) + d\left(\widetilde{\phi}_{s_{\max},\eta}\left(w_{\text{itr}}^{(T)}\right), \widetilde{\phi}_{s_{\max},0}\left(w_{\text{itr}}^{(T)}\right)\right)$$
$$\leq d\left(w_{\text{itr}}^{(T)}, \widetilde{\phi}_{s_{\max},\eta}\left(w_{\text{itr}}^{(T)}\right)\right) + \log C + \log(1+\eta)$$
$$\leq \frac{\delta^T \log \beta + \log C + \log \frac{1+\eta}{1-\eta}}{1-\delta} + \log C + \log(1+\eta)$$
$$\leq \frac{\delta^T \log \beta + 2\log C + 2\log(1+\eta) - \log(1-\eta)}{1-\delta}$$
$$\leq \frac{3\log C}{1-\delta},$$

where the first inequality follows from the triangle inequality of $d$, the second inequality applies Lemma C.5, the third inequality substitutes the previous bound, and the last inequality follows from the choice of $T$ (line 2 of Algorithm 1).

Set $\alpha := C^{3/(1-\delta)} = \Theta_{L,\theta,c}(1)$. By Fact 1, we have that $w^{(j_{\max})} = w_{\text{itr}}^{(T)}$ is $\alpha$-approximate at scale $s_{\max} = 2^{j_{\max}}$.

We now proceed with a downward recursion across scales (lines 9-12 of Algorithm 1). For each $j = j_{\max} - 1, \ldots, j_{\min}$, define $w^{(j)} := \widetilde{\phi}_{2^j, \eta}\big(w^{(j+1)}\big)$. By applying Lemma C.5 inductively, we obtain that

$$
\begin{aligned}
&d\left(w^{(j)}, \widetilde{\phi}_{2^j, \eta}\left(w^{(j)}\right)\right) \\
&= d\left(\widetilde{\phi}_{2^j, \eta}\left(w^{(j+1)}\right), \widetilde{\phi}_{2^j, \eta}^2\left(w^{(j+1)}\right)\right) \\
&\leq \delta \cdot d\left(w^{(j+1)}, \widetilde{\phi}_{2^j, \eta}\left(w^{(j+1)}\right)\right) + \log C + \log \frac{1+\eta}{1-\eta} \\
&= \delta \cdot \left(d\left(w^{(j+1)}, \widetilde{\phi}_{2^{j+1}, \eta}\left(w^{(j+1)}\right)\right) + \log 2\right) + \log C + \log \frac{1+\eta}{1-\eta} \\
&\leq \delta^{j_{\max}-j} \cdot \left(d\left(w^{(j_{\max})}, \widetilde{\phi}_{s_{\max}, \eta}\left(w^{(j_{\max})}\right)\right) + \log 2\right) + \sum_{i=0}^{j_{\max}-1-j} \delta^i \left(\log C + \log \frac{1+\eta}{1-\eta}\right) \\
&\leq \log \alpha + \log 2 + \frac{\log C + \log \frac{1+\eta}{1-\eta}}{1-\delta}.
\end{aligned}
$$

Using the above result, we can now verify that Equation (6) holds.

$$
\begin{aligned}
d\left(w^{(j)}, \phi_{2^j}\left(w^{(j)}\right)\right) &= d\left(w^{(j)}, \widetilde{\phi}_{2^j, \eta}\left(w^{(j)}\right)\right) + d\left(\widetilde{\phi}_{2^j, \eta}\left(w^{(j)}\right), \widetilde{\phi}_{2^j, 0}\left(w^{(j)}\right)\right) \\
&\leq d\left(w^{(j)}, \widetilde{\phi}_{2^j, \eta}\left(w^{(j)}\right)\right) + \log C + \log(1+\eta) \\
&\leq \log \alpha + \log 2 + \frac{\log C + \log \frac{1+\eta}{1-\eta}}{1-\delta} + \log C + \log(1+\eta) \\
&\leq \log \alpha + \log 2 + \frac{2\log C + 2\log(1+\eta) - \log(1-\eta)}{1-\delta} \\
&\leq 2\log \alpha + \log 2.
\end{aligned}
$$

And confirm that Equation (7) is satisfied.

$$
d\left(w^{(j+1)}, w^{(j)}\right) = d\left(w^{(j+1)}, \widetilde{\phi}_{2^j, \eta}\left(w^{(j+1)}\right)\right) = d\left(w^{(j+1)}, \widetilde{\phi}_{2^{j+1}, \eta}\left(w^{(j+1)}\right)\right) + \log 2 \leq 2\log 2\alpha.
$$

Therefore, $\mathcal{W} := \big\{w^{(j)} : j \in \mathcal{J} := \mathbb{Z} \cap [j_{\min}, j_{\max}]\big\}$ forms a $4\alpha^2$-approximate weight scheme. It remains to verify that the vector $z$ in line 13 of Algorithm 1 (see Proposition C.3 for details) is an $O(n \cdot |j_{\max} - j_{\min}|)$-overestimate with respect to $\mathcal{W}$.

For every $i \in [m]$, note that

$$
z_i = \frac{1}{1-\eta} \sum_{j \in \mathcal{J}} \widetilde{\sigma}_i\left(W_j^{1/2} A\right) \geq \sum_{j \in \mathcal{J}} \sigma_i\left(W_j^{1/2} A\right) \geq \max_{j \in \mathcal{J}} \sigma_i(W_j^{1/2} A),
$$

and

$$
\begin{aligned}
\|z\|_1 = \sum_{i \in [m]} z_i &= \frac{1}{1-\eta} \sum_{i \in [m]} \sum_{j \in \mathcal{J}} \widetilde{\sigma}_i\left(W_j^{1/2} A\right) \leq \frac{1+\eta}{1-\eta} \sum_{j \in \mathcal{J}} \sum_{i \in [m]} \sigma_i\left(W_j^{1/2} A\right) \\
&\leq \frac{1+\eta}{1-\eta} \sum_{j \in \mathcal{J}} \operatorname{Tr}(W_j^{1/2} A \left(A^\top W_j A\right)^+ (W_j^{1/2} A)^\top) \leq \frac{1+\eta}{1-\eta} \sum_{j \in \mathcal{J}} \operatorname{rank}(A) \\
&\leq \frac{1+\eta}{1-\eta} (j_{\max} - j_{\min}) n.
\end{aligned}
$$

## D. Proof of Theorem 3.5

*Proof of Theorem 3.5.* The construction follows the same three–step recipe as in the classical argument of Jambulapati et al. (2024): (1) pick anchor points $\hat{t}_i$ on each loss function; (2) use ModLevApprox to estimate leverage scores at those anchors

and convert them into weights; (3) add a quadratic bump to every $f_i$ to obtain a better-behaved family $\mathcal{F}^\circ$ together with an approximate weight vector $w^\circ$.

**Step 1: Anchor Selection.** Since each $f_i$ is $(L, \theta, c)$-proper, we can efficiently search over $t > 0$ to locate anchor points $\widehat{t}_i$ satisfying

$$\frac{1}{2} s_{\max} \le f_i(\widehat{t}_i) \le s_{\max}, \quad i \in [m],$$

as detailed in Proposition D.1. Thus, a quantum oracle $\mathcal{O}_t$ to these anchors can be constructed using $O(1)$ queries to $\mathcal{O}_{\mathcal{F}}$.

**Step 2: Quantum Leverage Score Approximation.** Invoke $\mathsf{ModLevApprox}(\mathcal{O}_A, \mathcal{O}_h, \frac{1}{2})$, where $\mathcal{O}_h$ queries the vector $h = (\widehat{t}_1^{-2}, \ldots, \widehat{t}_m^{-2})$, constructed in $\widetilde{O}(1)$ time via $O(1)$ queries to $\mathcal{O}_t$. The quantum algorithm uses $\widetilde{O}(r\sqrt{mn})$ queries to $\mathcal{O}_A$, $\widetilde{O}(\sqrt{mn})$ queries to $\mathcal{O}_{\mathcal{F}}$, and runs in $\widetilde{O}(r\sqrt{mn} + n^\omega + nr^2)$ time. It outputs an approximation $\widetilde{\sigma}$ satisfying $\frac{1}{2}\sigma(H^{1/2}A) \le \widetilde{\sigma}_i \le \frac{3}{2}\sigma(H^{1/2}A)$ for $H = \mathrm{diag}(h)$. Set

$$w_i^\circ := \delta/\widetilde{\sigma}_i, \quad i \in [m],$$

and define query access $\mathcal{O}_{w^\circ}$, where each query requires $O(r)$ queries to $\mathcal{O}_A$, $O(1)$ queries to $\mathcal{O}_{\mathcal{F}}$, and takes $\widetilde{O}(r)$ time.

**Step 3: Modified Loss Definition.** Define modified losses as:

$$f_i^\circ(t) = f_i(t) + s_{\max} w_i^\circ \cdot t^2, \quad \forall i \in [m].$$

Following Jambulapati et al. (2024, Fact 2.10), each $(f_i^\circ)^{1/2}$ is $\max\{1, L\}$-auto-Lipschitz, $\theta$-lower-homogeneous with constant $c$, making $\mathcal{F}^\circ$ a $(\max\{1, L\}, \theta, c)$-proper family. The correctness of Equation (4) and that $w^\circ$ is $\beta$-approximate at scale $s_{\max}$ directly follows from Jambulapati et al. (2024, Theorem 2.11), yielding $C_{\mathrm{INIT}} = 2(2L/c)^{2/\theta}$. $\qquad\square$

**Proposition D.1.** *Let $\mathcal{O}_f$ be an oracle for an $(L, \theta, c)$-proper function $f : \mathbb{R} \to \mathbb{R}_+$. For any $s_{\max} > s_{\min} > 0$, there exists an algorithm $\mathsf{FindSol}(\mathcal{O}_f, s_{\min}, s_{\max})$ that outputs $x > 0$ such that $s_{\min} \le f(x) \le s_{\max}$, making at most $O_{L,\theta,c}\left(\log \frac{s_{\max}}{s_{\min}}\right)$ queries to $\mathcal{O}_f$, and running in $O_{L,\theta,c}\left(\log \frac{s_{\max}}{s_{\min}}\right)$ time.*

*Proof of Proposition D.1.* Let $h = \sqrt{f}$. Recall Definition 1.2, $h$ is $L$-auto-Lipschitz and lower $\theta$-homogeneous with constant $c > 0$. We implement oracle access to $h$ via a single call to $\mathcal{O}_f$ and a square root, so the query complexity to $\mathcal{O}_h$ equals that to $\mathcal{O}_f$. Let $h_{\min} := \sqrt{s_{\min}}, h_{\max} := \sqrt{s_{\max}}$. Our goal is to find $x > 0$ such that $h_{\min} \le h(x) \le h_{\max}$.

We first query $h(1)$. If $h_{\min} \le h(1) \le h_{\max}$, we are done. Otherwise, we use exponential search to find an interval $[a, b]$ (with $b = 2a$ or $a = 2b$) such that $h$ crosses the range $[h_{\min}, h_{\max}]$ within this bracket.

**Case 1:** $h(1) < h_{\min}$. By lower $\theta$-homogeneity,

$$h(2^j) \ge c \cdot 2^{j\theta} h(1), \qquad j \ge 1.$$

Thus, for $j^\star = \left\lceil \frac{1}{\theta} \log_2 \frac{h_{\max}}{c \cdot h(1)} \right\rceil$, we have $h(2^{j^\star}) \ge h_{\max}$. Therefore, within $O(\frac{1}{\theta} \log \frac{h_{\max}}{c \cdot h(1)})$ queries we either find $x$ with $h(x) \in [h_{\min}, h_{\max}]$ or obtain a bracket $[2^{j^\star - 1}, 2^{j^\star}]$.

**Case 2:** $h(1) > h_{\max}$. For $j \ge 1$, taking $\lambda = 2^j$ and $x = 2^{-j}$ in the homogeneity property gives

$$h(1) = h(2^j \cdot 2^{-j}) \ge c \cdot 2^{j\theta} h(2^{-j}) \quad \Rightarrow \quad h(2^{-j}) \le \frac{h(1)}{c \cdot 2^{j\theta}}.$$

Hence for $j^\star = \left\lceil \frac{1}{\theta} \log_2 \frac{h(1)}{c \cdot h_{\min}} \right\rceil$, we have $h(2^{-j^\star}) \le h_{\min}$, and again we either hit the target range or obtain a bracket $[2^{-j^\star}, 2^{-(j^\star - 1)}]$.

**Bisection step.** Assume w.l.o.g. we have a bracket $[a, b]$ with $b = 2a$ and $h(a) \le h_{\min} < h_{\max} \le h(b)$ (the mirrored case is analogous). Let $\Delta := h_{\max} - h_{\min} > 0$ and let the half-length be $\delta_0 := \frac{b-a}{2} = \frac{a}{2}$. For $t \ge 0$, let the half-length after $t$ bisections be $\delta_t := \delta_0/2^t$. By lower $\theta$-homogeneity (applied with $x = \delta_{t+1}$ and $\lambda = 2$),

$$h(\delta_{t+1}) \le \frac{1}{c2^\theta} h(\delta_t) \quad \Rightarrow \quad h(\delta_t) \le \left(\frac{1}{c2^\theta}\right)^t h(\delta_0).$$

Moreover, since $b - a = a$ and $h(a) \leq h_{\min}$, we have

$$h(\delta_0) = h\left(\frac{a}{2}\right) \leq \frac{1}{c2^\theta} h(a) \leq \frac{h_{\min}}{c2^\theta}.$$

Combining the two displays yields

$$Lh(\delta_t) \leq L\left(\frac{1}{c2^\theta}\right)^{t+1} h_{\min}.$$

Choose

$$t \geq \left\lceil \frac{1}{\theta + \log_2 c} \log_2\left(\frac{Lh_{\min}}{\Delta}\right) \right\rceil - 1.$$

Then $Lh(\delta_t) \leq \Delta$. Let $m$ be the midpoint of the current bracket $[a', b']$ (of length $\delta_{t-1}$). By the $L$-auto-Lipschitz property,

$$h(m) \leq h(a') + Lh(\delta_t) \leq h_{\min} + \Delta = h_{\max},$$
$$h(m) \geq h(b') - Lh(\delta_t) \geq h_{\max} - \Delta = h_{\min},$$

so $h(m) \in [h_{\min}, h_{\max}]$ and the midpoint succeeds.

Overall, the total query complexity of FindSol is $O_{L,\theta,c}\left(\log \frac{s_{\max}}{s_{\min}}\right)$, under the assumption that $f(1)$ is bounded by $\text{poly}(s_{\min}, s_{\max})$. The time is linear in the number of queries.

$\square$

## E. Proof of Theorem 4.1

Before proving Theorem 4.1, we restate two quantum algorithms that are invoked as subroutines in our algorithm.

**Theorem E.1** (Multiple Quantum State Preparation, Hamoudi (2022, Theorem 1))**.** *Let $1 \leq k \leq n$. There exists a quantum algorithm* MultiSample$(\mathcal{O}_w, k)$ *that, given query access $\mathcal{O}_w$ to a nonnegative vector $w \in \mathbb{R}_+^n$, outputs with high probability a sample sequence $\vartheta \in [n]^k$ in which each element $i$ is drawn with probability proportional to $w_i$. The algorithm makes $\widetilde{O}(\sqrt{kn})$ queries to $\mathcal{O}_w$ and runs in $\widetilde{O}(\sqrt{kn})$ time.*

**Theorem E.2** (Quantum Sum Estimation, Li et al. (2019, Lemma 3.1))**.** *There exists a quantum algorithm* SumEstimate$(\mathcal{O}_w, \epsilon)$ *that, given query access $\mathcal{O}_w$ to a nonnegative vector $w \in \mathbb{R}_+^n$, outputs with high probability an $\epsilon$-approximation $\widetilde{s}$ to the sum $s = \sum_{i \in [n]} w_i$. The algorithm uses $\widetilde{O}(\sqrt{n}/\epsilon)$ queries to $\mathcal{O}_w$ and runs in $\widetilde{O}(\sqrt{n}/\epsilon)$ time.*

We will use the following result from Jambulapati et al. (2024) in a black-box manner.

**Theorem E.3.** *Let $F(x) := \sum_{i=1}^m f_i(\langle a_i, x \rangle)$ be a total loss function defined by vectors $a_1, \ldots, a_m \in \mathbb{R}^n$ and an $(L, \theta, c)$-proper loss functions family $\mathcal{F} = \{f_1, \ldots, f_m\}$. Let $0 \leq s_{\min} < s_{\max}$. Given any $O_{L,\theta,c}(1)$-approximate weight scheme $\mathcal{W}$ with respect to matrix $A$ (with rows $a_i$), loss family $\mathcal{F}$, and interval $\mathcal{J} := [\log_2 s_{\min} - 4 \log m, \log_2 s_{\max}] \bigcap \mathbb{Z}$, with high probability, importance sampling each function $f_i \in \mathcal{F}$ with probability*

$$p_i \geq \max_{j \in \mathcal{J}} \sigma_i(W_j^{1/2} A)$$

*yields an $s$-sparse $\epsilon$-approximate sparsifier of $F$ over the range $[s_{\min}, s_{\max}]$, where*

$$s = O(\|p\|_1 \log^3 m / \epsilon^2).$$

*Proof of Theorem 4.1.* According to Theorem 3.5, we obtain query access to $\mathcal{F}^\circ$ and $w^\circ$ in $\widetilde{O}(r\sqrt{mn} + n^\omega + nr^2)$ time, and each query costs $\widetilde{O}(r)$ time. These satisfy the property that $w^\circ$ is an $O(m^4 s_{\max}/(\epsilon s_{\min}))$-approximate weight at scale $s_{\max}$ with respect to $\mathcal{F}^\circ$, and moreover $\mathcal{F}^\circ$ satisfies the condition

$$\sum_{i=1}^m f_i(\langle a_i, x \rangle) \leq s_{\max} \quad \Rightarrow \quad 0 \leq \sum_{i=1}^m f_i^\circ(\langle a_i, x \rangle) - f_i(\langle a_i, x \rangle) \leq \frac{C_{\text{INIT}} \epsilon s_{\min}}{m}. \tag{9}$$

Note that $|\mathcal{J}| = j_{\max} - j_{\min} = O(\log(m s_{\max}/s_{\min})) = \widetilde{O}(\log(s_{\max}/s_{\min}))$. By applying Theorem 3.4, we obtain query access to a multiscale $O(n \log(s_{\max}/s_{\min}))$-overestimates $z$ with respect to $\mathcal{F}^{\circ}$. This construction requires $\widetilde{O}\left((r\sqrt{mn} + n^{\omega} + nr^2)\log(s_{\max}/s_{\min})\right)$ time, and each query can be executed in $\widetilde{O}(r\log(s_{\max}/s_{\min}))$ time. Combining this with the multiple quantum state preparation (Theorem E.1), we can generate a sequence $\vartheta \in [m]^M$ in $\widetilde{O}(\sqrt{Mm} \cdot \log(s_{\max}/s_{\min})) = \widetilde{O}(r\sqrt{mn}\log(s_{\max}/s_{\min})/\epsilon)$ time. The quantum sum estimation procedure (Theorem E.2) then runs in $\widetilde{O}(r\sqrt{m}\log(s_{\max}/s_{\min}))$ time, producing $\widetilde{\nu}$ as a 0.1-approximation of $\nu = \|z\|_1$. Therefore, the overall time complexity is $\widetilde{O}\left((n^{\omega} + nr^2 + r\sqrt{mn}/\epsilon) \cdot \log(s_{\max}/s_{\min})\right)$.

It remains to show that

$$\widetilde{F}(x) := \sum_{i=1}^{m} \frac{\#\{k \in [M] : \vartheta_k = i\}}{1.1 M z_i / \widetilde{\nu}} f_i(\langle a_i, x \rangle)$$

is an $\widetilde{O}(\log(s_{\max}/s_{\min}) \cdot n/\epsilon^2)$-sparse $\epsilon$-approximate sparsifier of $F$ over the range $[s_{\min}, s_{\max}]$.

In each sampling round (out of $M$ total rounds), the probability of selecting each $f_i \in \mathcal{F}$ can be expressed (after reweighting) as

$$1.1 z_i \cdot \frac{\nu}{\widetilde{\nu}} \geq \sigma_i^{\mathrm{ms}}(A, \mathcal{W}^{\circ}) \geq \max_{j \in \mathcal{J}} \sigma(W^{\circ}{}_j^{1/2} A),$$

where $W^{\circ}$ is an $O(1)$-approximate weight scheme with respect to $\mathcal{F}^{\circ}$. The inequalities follow from the bound $\widetilde{\nu} \leq 1.1\nu$ and from Definition 3.3.

Applying Theorem E.3, we obtain that $\widetilde{F}(x)$ is an $O(\|z\|_1 \log^3 m/\epsilon) = \widetilde{O}(n \log(s_{\max}/s_{\min})/\epsilon^2)$-sparse $\epsilon$-approximate sparsifier of $F^{\circ}$ over the range $[s_{\min}, s_{\max}]$. Finally, by Equation (9), for all $x$ such that $s_{\min} \leq F(x) \leq s_{\max}$, we have

$$
\begin{aligned}
\left| \widetilde{F}(x) - F(x) \right| &\leq \left| \widetilde{F}(x) - F^{\circ}(x) \right| + |F^{\circ}(x) - F(x)| \\
&\overset{Equation\ (9)}{\leq} \epsilon F^{\circ}(x) + \frac{C_{\mathrm{INIT}} \epsilon s_{\min}}{m} \\
&\overset{Equation\ (9)}{\leq} \epsilon \left( F(x) + \frac{C_{\mathrm{INIT}} \epsilon s_{\min}}{m} \right) + \frac{C_{\mathrm{INIT}} \epsilon s_{\min}}{m} \\
&\leq \epsilon F(x) + \frac{C_{\mathrm{INIT}}(\epsilon^2 + \epsilon)}{m} F(x) \\
&\leq 2\epsilon F(x).
\end{aligned}
$$

By setting $\epsilon \leftarrow \frac{\epsilon}{2}$, we obtain what we desired. $\qquad\square$

