# OpenReview forum: "Accelerating Regression Tasks with Quantum Algorithms"
_ICML.cc/2026/Conference — ICML 2026 regular_

### Official Review · Reviewer_MNzx · 2026-03-09

**Soundness:** 3
**Presentation:** 2
**Significance:** 3
**Originality:** 3
**Overall Recommendation:** 4
**Confidence:** 3

**Summary:**

This paper presents a unified quantum algorithmic framework for accelerating a broad class of regression tasks, including linear, multiple, Lasso, Ridge, Huber, and $\ell_p$-type regressions. The authors achieve this by adapting the Generalized Linear Model (GLM) sparsification framework—a recent classical breakthrough—into the quantum domain. The core contribution is a quantum algorithm that efficiently constructs an $\epsilon$-approximate sparsifier (a much smaller, weighted subset of the dataset that preserves the objective value) by using quantum leverage score approximations to initialize and refine "Multiscale Leverage Score Overestimates" (MLSO). By integrating this with multiple quantum state preparation and quantum sum estimation, the algorithm achieves a runtime of $\tilde{O}(r\sqrt{mn}/\epsilon)$, offering up to a quadratic speedup in the number of samples ($m$) over the best classical algorithms.

**Compliance With Llm Reviewing Policy:**

Affirmed.

**Key Questions For Authors:**

The paper is exceptionally dense and mathematically heavy. Can you provide flowcharts, diagrams, or numerical simulations that could help the reader build an intuition for the algorithmic pipeline or estimate the actual size of the hidden constants.

**Limitations:**

yes

**Strengths And Weaknesses:**

Strengths:

The paper makes a significant theoretical contribution to the quantum algorithms literature by unifying a wide variety of regression problems under a single quantum sparsification framework. Achieving a $\sqrt{m}$ speedup for complex robust regressions is mathematically rigorous and deeply knowledgeable.

The authors demonstrate an grasp of state-of-the-art classical randomized numerical linear algebra, successfully quantizing a very recent classical breakthrough from STOC 2024. The theoretical pipeline is logically sound, and the proofs in the appendices are thorough.


Weaknesses:
The algorithm's sublinear time complexity fundamentally relies on the existence of a Quantum Random Access Memory (QRAM). While QRAM is a standard assumption in theoretical quantum query complexity, it is widely believed to be extremely expensive and engineering-prohibitive in the near term.

To guarantee a worst-case $(1 \pm \epsilon)$ relative error without reverting to uniform sampling (which fails on coherent matrices), the algorithm relies heavily on quantum leverage score approximation (Theorem 2.1). This subroutine introduces a brutal $O(\epsilon^{-4})$ time complexity dependence. In practical machine learning tasks where high precision (small $\epsilon$) is required, an $\epsilon^{-4}$ overhead quickly negates the $\sqrt{m}$ quantum speedup, severely restricting the algorithm's practical utility.

The use of $\tilde{O}$ notation hides potentially massive polylogarithmic factors, conditioning numbers, and constants (e.g., $C_{INIT}$). Because of these hidden overheads and the $\epsilon^{-4}$ bottleneck, the theoretical "quadratic speedup" might only manifest for astronomically large datasets.

---

> ### Author Rebuttal · Authors · 2026-03-30
>
> We thank the reviewer for the positive assessment and for the very helpful suggestions on clarity and presentation. We address the reviewer’s main concerns in turn.
>
> 1 . **On the $\epsilon$-dependence and hidden complexity terms.**
>
> We would like to clarify one important point regarding the dependence on $\epsilon$. The $\epsilon^{-4}$ factor appears in the leverage-score approximation subroutine (Theorem 2.1 / Proposition B.1), where $\epsilon$ is the *internal accuracy parameter* of ModLevApprox. This is **not** the same as the target sparsification error parameter in Theorem 4.1. In our algorithms, ModLevApprox is invoked only with **constant internal accuracy**---specifically, $\epsilon = 0.1$ in Algorithm 1 and $\epsilon = 1/2$ in the initialization procedure of Theorem 3.5---so the resulting cost contributes only to the $\widetilde{O}(r\sqrt{mn} + n^\omega + nr^2)$ part of the runtime. Accordingly, the final sparsification algorithm in Theorem 4.1 has runtime $\widetilde{O}((n^\omega + nr^2 + r\sqrt{mn}/\epsilon)\log(s_{\max}/s_{\min}))$, rather than $\widetilde{O}(\epsilon^{-4})$. Moreover, Remark 4.3 shows that the dependence on $m$ and $1/\epsilon$ is **near-optimal** in the relevant sense. We do agree, however, that $\widetilde{O}$ notation may hide nontrivial polylogarithmic factors and constants, so the crossover point at which the asymptotic speedup becomes visible may still depend on the problem regime. We will make this interpretation more explicit in the final version.
>
> 2 . **On QRAM and the interpretation of the speedup.**
>
> We do agree that the QRAM assumption limits the near-term practical interpretation of the result. Our paper works in the standard quantum query model stated in Section 2, and Appendix A explicitly notes that QRAM is a standard theoretical abstraction but remains an engineering challenge. Thus, our claims should be interpreted primarily as a theoretical quantum speedup under standard query-model assumptions, rather than as a claim of immediate practical advantage.
>
> 3 . **On exposition and empirical intuition.**
>
> We also appreciate the suggestion on exposition. In the final version, we will add a compact figure illustrating the logic of Algorithms 1 and 2, together with a brief summary of the main hidden terms in the complexity bounds. We also note that meaningful empirical evaluation is difficult in our setting, since the algorithm is formulated in the quantum query model and relies on primitives such as quantum query access, QRAM, multiple quantum state preparation, and quantum sum estimation, which are currently difficult to implement faithfully on existing hardware or to simulate at meaningful classical scales. For this reason, our focus in the present paper is on the theoretical framework and complexity guarantees. Nevertheless, we agree that clearer visual summaries and a more explicit discussion of the complexity terms would make the algorithmic pipeline much easier to follow.
>
> We thank the reviewer again for the careful reading and constructive suggestions. We hope the clarifications above help address the concerns more clearly, and we would be happy to provide further clarification if helpful.

---

> > ### Author Rebuttal · Reviewer_MNzx · 2026-04-05
> >
> > I thank the authors for the precise clarification regarding the internal accuracy parameter, which successfully resolves my main concern about the $\epsilon^{-3}$ scaling bottleneck. Given the solid theoretical foundation and the commitment to explicitly detailing the hidden constants and algorithmic flowcharts, I am confident in maintaining my positive score.

---

### Official Review · Reviewer_xCcm · 2026-03-10

**Soundness:** 3
**Presentation:** 2
**Significance:** 3
**Originality:** 3
**Overall Recommendation:** 4
**Confidence:** 1

**Summary:**

This paper provides a unified quantum framework that uses generalized linear model (GLM) sparsification framework to accelerate various regression tasks. The approach introduces a quantum method to compute multiscale leverage score overestimation (MLSO) and combines it with quantum multiple sampling and summation estimation, resulting in improvements in both time and efficiency.

**Compliance With Llm Reviewing Policy:**

Affirmed.

**Key Questions For Authors:**

1. The claimed speedup appears to rely on strong assumptions about data access, such as quantum oracle/superposition access and the availability of quantum random access memory (QRAM). Under what most realistic access patterns would your results still yield a meaningful advantage? How would the overall complexity change if these assumptions were weakened?
2. The access model assumes efficient quantum procedures for row queries and nonzero index retrieval; can you specify the preprocessing requirements (classical time/memory) needed to enable these oracles, and whether these preprocessing costs erase the claimed asymptotic gain in realistic settings?

**Limitations:**

While the paper is primarily theoretical and presents no immediate societal risks, the current discussion does not sufficiently articulate the practical limitations critical for a machine learning audience. The authors should more explicitly address the stringency of their data-access and QRAM-like assumptions, clarifying how these requirements may preclude near-term speedups in practical ML pipelines. Furthermore, the manuscript would benefit from a rigorous regime analysis, identifying specific parameter ranges where hidden polynomial factors—such as those in $n$ and $1/\varepsilon$, or potential $n^\omega$ costs—might dominate and negate the advertised $\sqrt{m}$ advantage.

**Strengths And Weaknesses:**

Strengths: this paper integrates multiple regression objectives into a unified sparse-based framework, providing clear inferences and coverage. It achieves a quantum speedup of $\sqrt{m}$ relative to the primary sample size parameter, improving or surpassing previous quantum results for linear and ridge regression within the relevant parameter range. Additionally, the paper presents an explicit sparsification process and complexity characterization, enabling the proposed acceleration mechanism to trace back from leverage/sensitivity estimation and weighted sampling to simplified optimization instances.

Weaknesses: Larger hidden terms (e.g., polynomial factors in n and 1/ε, as well as potential n^ω-type costs) may dominate the runtime and obscure the claimed m improvement. However, the paper does not provide sufficiently clear state diagrams to describe when the acceleration truly materializes. And the paper's presentation is challenging for non-experts and lacks numerical examples or empirical evidence, making it difficult to determine the significance of these bounds for typical problem sizes.

---

> ### Author Rebuttal · Authors · 2026-03-30
>
> We thank the reviewer for the thoughtful questions and constructive suggestions. We address the reviewer’s main concerns in turn.
>
> 1 . **On the access model and the scope of our result.**
>
> Our results are proved in the standard quantum query model of Section 2, which assumes quantum query access to the input matrix (including both entry queries and nonzero-index queries), oracle access to the loss-function family, and access to a quantum-read/classical-write RAM (QRAM). The reported complexity already counts elementary gates, quantum queries, and QRAM operations. Accordingly, our theorem should be interpreted as a conditional quantum speedup under these standard access assumptions, rather than as an unconditional end-to-end speedup from an arbitrary classical data representation.
>
> 2 . **On what is and is not included in the complexity bound.**
>
> The complexity bound in Theorem 4.1 does not separately account for the cost of building the required quantum access structure from a raw classically stored dataset; this setup is part of the assumed input model. Therefore, if one starts from an ordinary sequential-access classical matrix and must construct the required row/entry/nonzero-index access from scratch for a single regression instance, that setup cost can already be at least linear in the input size and may erase the asymptotic gain. For this reason, the result is most meaningful in settings where the required access is already available, or where the same dataset is reused across many regression tasks so that the setup cost can be amortized. We will make this scope and limitation more explicit in the final version. As noted in Appendix A, if one focuses only on quantum query complexity, the QRAM assumption can be removed at the cost of a polynomial increase in time complexity.
>
> 3 . **On the parameter regime in which the $m$-improvement is visible.**
>
> Under the above access model, the formal runtime of our sparsification algorithm is
> $$
> \widetilde O\left((n^\omega + nr^2 + r\sqrt{mn}/\epsilon)\log(s_{\max}/s_{\min})\right).
> $$
> As discussed in the paper, a natural sparsification regime is $\epsilon=\Omega(\sqrt{n/m})$, in which the sparsifier is smaller than the original dataset and the leading $m$-dependent term $r\sqrt{mn}/\epsilon$ is strictly smaller than the classical $mr$. Thus, the claimed speedup should be understood as applying in parameter regimes where this leading $m$-dependent term is dominant, rather than uniformly across all settings. We also agree that the current draft should state this more explicitly. In the final version, we will add a short regime discussion after Theorem 4.1 and expand the limitations discussion to clarify when the $n^\omega$, $nr^2$, and $1/\epsilon$ terms may dominate.
>
> We also agree that the presentation could better highlight these tradeoffs. Since the paper is formulated in the quantum query model, we do not include numerical experiments; however, we will add a compact schematic of the pipeline and make the practical scope and parameter regimes more explicit in the final version.
>
> We thank the reviewer again for the careful reading and constructive suggestions. We would be happy to clarify any further questions.

---

> > ### Author Rebuttal · Reviewer_xCcm · 2026-04-02
> >
> > Thanks for your reply.

---

### Official Review · Reviewer_vBot · 2026-03-12

**Soundness:** 3
**Presentation:** 3
**Significance:** 2
**Originality:** 2
**Overall Recommendation:** 3
**Confidence:** 3

**Summary:**

This paper presents a quantum framework for "accelerating" a class of regression tasks via Generalized Linear Model (GLM) sparsification. This is done by "quantizing" the recent classical GLM sparsification of Jambulapati et al. (2024). The quantized algorithm invovles: (1) construct initial approximate weights via quantum leverage score approximation, (2) refine them into Quantum Multiscale Leverage Score Overestimates (QMLSO), (3) performs efficient importance sampling via Multiple Quantum State Preparation (Hamoudi, 2022) and Quantum Sum Estimation (Li et al., 2019), resulting in a sparse reweighted subset (Algorithm 2). For an $m \times n$ matrix $A$ with row sparsity $r$, the main result (Theorem 4.1) achieves $\tilde{O}(r\sqrt{mn}/\epsilon + \text{poly}(n,1/\epsilon))$ quantum time for producing an $\epsilon$-approximate sparsifier. Compared to the classical $\tilde{O}(mr + \text{poly}(n,1/\epsilon))$, this yields up to a quadratic speedup in $m$. Then the framework is applied to linear, multiple, ridge, Lasso, Huber, $\ell_p$, and $\gamma_p$ regression.

**Compliance With Llm Reviewing Policy:**

Affirmed.

**Final Justification:**

I appreciate the authors' detailed response, and am raising the score to three. Reason for not higher is the high likelihood of being dequantized, which degrades the value of this type of speed up.

**Key Questions For Authors:**

* What new technical challenges were introduced in quantifying the convergence guarantees from Jambulapati et al. (2024)?
* I would appreciate the authors' comment on the possibility of this work being dequantized via quantum-inspired classical algorithms.
* On a similar vein, could this work avoid the reliance on QRAM?

**Limitations:**

Yes

**Strengths And Weaknesses:**

### Strengths
* The quantum algorithm from this paper covers a range of regression tasks under a single framework via $(L,\theta,c)$-proper loss functions. This is in contrast to prior works (Song et al., 2023; Apers & Gribling, 2024) which addressed only linear regression and spectral sparsification.
* The $\tilde{O}(\sqrt{mn})$ row-query dependence matches the $\Omega(\sqrt{mn})$ lower bound of Apers & Gribling (2024) for constant $\epsilon$, and the $\tilde{O}(\sqrt{mn}/\epsilon)$ dependence matches the $\Omega(\sqrt{mn}/\epsilon)$ lower bound of Apers & de Wolf (2022) in the regime $m = O(n^2)$.
* The classical algorithm from Jambulapati et al. (2024) requires explicit weight vectors and iterative updates. Adapting this to the quantum setting seems technically non-trivial.

### Weaknesses
* The algorithmic framework, including correctness arguments, closely follows Jambulapati et al. (2024), and the key quantum subroutine (leverage score approximation) is from Apers & Gribling (2024).
* More significantly, this work relies on the QRAM, which is controverisial whether or not it can be implemented efficiently in practice. Moreover, quantum algorithms that rely on QRAM can often be "dequantized" via quantum-inspired classical algorithms with appropriate sample/query access, even in the case of exponential separation (HHL algorithm). Considering this aspect, the (at most quadratic) speedup achieved in this work seems not significant enough.

---

> ### Author Rebuttal · Authors · 2026-03-30
>
> We thank the reviewer for the thoughtful comments. We address the three questions in turn.
>
> 1 . **On the technical challenges.**
>
> Our contribution is not a black-box substitution of the leverage-score routine of Apers \& Gribling into the classical framework of Jambulapati et al. The main technical challenge is to quantize the *entire* multiscale GLM sparsification pipeline while preserving its convergence guarantees. More concretely, in the general proper-loss setting, one cannot simply plug a quantum leverage-score primitive into the classical algorithm and obtain the desired result. Our framework introduces MLSO/QMLSO as quantum-accessible analogues of the GLM importance scores, proves that the iterative refinement remains valid when leverage scores are only approximately computed, and develops a fully quantum initialization procedure that constructs both an initial approximate weight vector and a modified loss family so that the subsequent refinement can proceed entirely in the quantum setting. In particular, the noisy contraction analysis in Appendix B is precisely the part that does not follow automatically from prior work: it is what makes the multiscale refinement robust to the approximation errors introduced by the quantum subroutines.
>
> 2 . **On the possibility of dequantization.**
>
> We agree that dequantization is an interesting direction. However, our paper is concerned with establishing a quantum speedup for the multiscale GLM sparsification pipeline in the standard quantum query model, starting from the classical GLM sparsification framework and developing its quantum counterpart. We do not address in this paper whether comparable guarantees can also be achieved by a quantum-inspired classical algorithm for such general proper-loss settings, and we leave this question for future investigation.
>
> 3 . **On the reliance on QRAM.**
>
> We agree that the reliance on QRAM is an important practical limitation. Our result should be interpreted as a theoretical quantum speedup in the standard quantum query model, where QRAM is a standard abstraction used to formalize efficient quantum access to data. As we note in the appendix, scalable realization of QRAM remains challenging in practice. At present, our framework relies on this access model in an essential way. Removing the QRAM assumption while retaining comparable guarantees would likely require substantially new ideas, rather than a minor modification of the current approach. We will make this practical limitation and the scope of our claim more explicit in the final version.
>
> We thank the reviewer again for the careful reading and helpful comments. We would be happy to clarify any further questions.

---

> > ### Author Rebuttal · Reviewer_vBot · 2026-04-03
> >
> > Thank you for the response.

---

### Official Review · Reviewer_bQwS · 2026-03-12

**Soundness:** 3
**Presentation:** 2
**Significance:** 2
**Originality:** 2
**Overall Recommendation:** 3
**Confidence:** 2

**Summary:**

This paper considers quantum methods to accelerate a broad class of regression tasks.
The paper first provide a novel quantum algorithm for multiscale leverage score overestimation and then apply this novel quantum algorithm in the generalized linear model sparsification.
The proposed quantum method enjoys a time complexity of $\tilde{O}(n^{\omega}+nr^2 + r\sqrt{mn}/\epsilon)$, which is better than the prior classical and quantum algorithms.

**Compliance With Llm Reviewing Policy:**

Affirmed.

**Final Justification:**

I appreciate the authors' detailed response. I decide to keep my overall assessment unchanged since I'm not sure about the significance compared to the prior work Song et al, 2024. I am not an expert in this field and do not have strong objection on this paper.

**Key Questions For Authors:**

1. Please discuss how your QMLSO algorithm compared to the classical algorithms on the MLSO tasks. Is the quantum speedup in Section 4 mainly comes from QMLSO in Theorem 3.4?

2. Please discuss your technical difference with Song et al., 2023 and the concurrent work Song et al., 2025. The quantum speedup in this work upon the prior methods seems mainly on improving the $\sqrt{m}n^{1.5}/\epsilon$ to $r\sqrt{mn}/\epsilon$ by considering the sparsity, which seems rather limited.

**Limitations:**

Yes.

**Strengths And Weaknesses:**

**Strength**
1. This paper considers an important and fundamental probelem of generalized linear regression.
2. This paper provides novel quantum algorithms on multiscale leverage score estimates and apply it to the generalized linear models sparsification.
3. The time complexity given in this paper is better than the prior classical and quantum algorithms.

**Weakness**
Please refer to the question part.

---

> ### Author Rebuttal · Authors · 2026-03-30
>
> We sincerely thank the reviewer for the thoughtful and detailed comments. We address the questions below in turn.
>
> 1 . **On the source of the speedup.**
>
> For the first question, we would like to clarify that the quantum speedup in Section 4 does not arise from a single isolated subroutine, but from the combination of two key ingredients. First, via WeightInitialize and QMLSO, we construct a quantum data structure that supports efficient queries to the MLSO values after only sublinear-time preprocessing, without explicitly outputting all $m$ importance scores; this is crucial, since explicitly writing down the full score vector already requires at least linear time in $m$. Second, building on this query access, we use Multiple Quantum State Preparation together with Quantum Sum Estimation to perform the downstream importance sampling and reweighting efficiently. Therefore, in the general GLM setting, QMLSO in Theorem 3.4 is indeed the core algorithmic contribution of our framework: it quantizes the construction of multiscale weights/importance scores in the classical framework, and its output is not an explicit score vector, but a queryable quantum data structure that can be used directly in the later sampling stage, whereas its classical counterpart typically requires $\widetilde{O}(mr)$ time. At the same time, to make the entire pipeline fully quantum, we also develop the initialization procedure in Theorem 3.5, which generates a suitable initial weight vector by slightly modifying the loss functions and thereby enables the subsequent iterative refinement. In this sense, QMLSO is the key enabling module in Section 4, but the leading improvement in the $m$-dependence is achieved jointly by the initialization/QMLSO stage and the downstream quantum sampling/reweighting stage, rather than by Theorem 3.4 alone.
>
>
>
> 2 . **On the linear-regression special case and relation to prior work.**
>
> As noted in Remark 4.2, in the special case of linear regression our framework does not require the full multiscale leverage score overestimate (MLSO) machinery. Concretely, in Algorithm 2, Steps 2--5 can be replaced by the quantum leverage-score routine of Apers and Gribling (2023). Once such a spectral sparsifier is constructed, linear regression and its standard variants (such as multiple and ridge regression) can then be solved on top of that sparsifier using standard techniques. Song et al. (2023) can be viewed as essentially following the same spectral-sparsification route. Thus, for linear regression and its variants, the improvement should indeed be understood as a refinement of that framework, where our analysis preserves the row-sparsity parameter $r$, rather than pessimistically replacing it by $n$.
>
> That said, the restricted linear case is not the main contribution of the paper. Our primary technical contribution is to move beyond the homogeneous quadratic setting and develop a unified quantum GLM sparsification framework for a broad class of losses, including $\ell_p$, Huber, Lasso, and especially the non-homogeneous $\gamma_p$ family. In these general settings, one genuinely needs the MLSO formalism, the QMLSO refinement procedure in Theorem 3.4, and the fully quantum initialization routine in Theorem 3.5 to quantize the contractive GLM-sparsification framework of Jambulapati et al. These ingredients enable our method to handle the multiscale structure that is absent from ordinary linear regression. Thus, the characterization that our improvement is “rather limited” applies only to the linear-regression special case, not to the general proper-loss setting addressed in this work.
>
> Regarding the concurrent work of Song et al. (2025), the technical route is also different. They study $\ell_p$ regression within a quantum sensitivity-sampling framework and present a general quantum coreset framework based on recursively estimating sensitivities and resampling from those estimates; for $\ell_p$ regression, they obtain an $\epsilon$-coreset construction in time $\widetilde{O}\left(n\sqrt{mn}(\epsilon^{-3}+\sqrt{n})\right)$. By contrast, our work starts from the GLM sparsification framework and yields $\widetilde{O}(r\sqrt{mn}/\epsilon + nr^2 + n^{\omega})$ quantum sparsification for proper losses, giving **strictly better** parameter dependence. Finally, as discussed in Remark 4.3, the dependence on $1/\epsilon$ in our algorithm is **near-optimal** up to logarithmic factors.
>
> We thank the reviewer again for the thoughtful questions. If there are any further questions or points that would benefit from additional clarification, we would be very happy to elaborate.

---

> > ### Author Rebuttal · Reviewer_bQwS · 2026-04-01
> >
> > Thanks for your response to my questions.

---

> > > ### Author Response · Authors · 2026-04-03
> > >
> > > Thank you for reviewing our response and confirming that your concerns are fully resolved. In light of this, we would be very grateful if you could consider adjusting the score to reflect this positive resolution. We truly appreciate your time and support.

---

### Decision · Program_Chairs · 2026-04-30

**Decision:**

Accept (regular)

**Comment:**

All the reviewers agreed that the paper's theoretical contribution is interesting and non-trivial. It is able to solve multiple regression objectives such as least squares, lasso, ridge, Huber and lp, each with a quadratic speedup in the sample size. The main idea is to use the recent sparsification framework of Jambulapati et al. On the negative side, reviewers were considered about the heavy reliance on quantum RAM, which we are far from realizing in practice, as well as potentially large polylogarithmic and condition number dependencies. Still, the paper offers a useful and technically solid theoretical contribution to the quantum ML/optimization literature, which warrants acceptance.